# PHANTOM: A Benchmark for Hallucination Detection in Financial Long-Context QA

**Lanlan Ji    Dominic Seyler    Gunkirat Kaur**
**Manjunath Hegde    Koustuv Dasgupta    Bing Xiang**
Goldman Sachs
{lanlan.ji, dominic.seyler, gunkirat.x.kaur}@gs.com
{manjunath.y.hegde, koustuv.x.dasgupta, bing.xiang}@gs.com

## Abstract

While Large Language Models (LLMs) show great promise, their tendencies to hallucinate pose significant risks in high-stakes domains like finance, especially when used for regulatory reporting and decision-making. Existing hallucination detection benchmarks fail to capture the complexities of financial benchmarks, which require high numerical precision, nuanced understanding of the language of finance, and ability to handle long-context documents. To address this, we introduce PHANTOM, a novel benchmark dataset for evaluating hallucination detection in long-context financial QA. Our approach first generates a seed dataset of high-quality "query-answer-document (chunk)" triplets, with either hallucinated or correct answers - that are validated by human annotators and subsequently expanded to capture various context lengths and information placements. We demonstrate how PHANTOM allows fair comparison of hallucination detection models and provides insights into LLM performance, offering a valuable resource for improving hallucination detection in financial applications. Further, our benchmarking results highlight the severe challenges out-of-the-box models face in detecting real-world hallucinations on long context data, and establish some promising directions towards alleviating these challenges, by fine-tuning open-source LLMs using PHANTOM.[1]

## 1   Introduction

Large Language Models(LLMs) have demonstrated remarkable capabilities across a wide range of natural language tasks (Brown et al., 2020; Zhao et al., 2023; Gemini Team et al., 2023; Achiam et al., 2023). The adoption of LLMs in specialized domain like finance presents exciting opportunities, but also introduces the risk of hallucinations – outputs that are factually incorrect, inconsistent with source or misleading (Huang et al., 2025; Ji et al., 2023; Zhang et al., 2023; Das et al., 2023). While hallucinations can be problematic in any domain, their consequences in finance can be especially damaging, given that common finance AI use cases involve interpreting and presenting important financial data and documents. Although research on hallucination detection methods is growing, a significant gap exists in evaluating these methods specifically on financial data (Kang and Liu, 2023). Financial documents possess distinct characteristics, including high numerical precision requirements, domain-specific terminology, and intricate relationships between data points. These characteristics necessitate specialized evaluation benchmarks (Choi et al., 2025). The scarcity of benchmark datasets specifically designed for evaluating hallucination detection methods within the financial domain presents a significant obstacle to advancing research in this critical area (Kang and Liu, 2023).

---

[1]Dataset available at: `huggingface.co/datasets/seyled/Phantom_Hallucination_Detection`

39th Conference on Neural Information Processing Systems (NeurIPS 2025) Track on Datasets and Benchmarks.

Furthermore, financial documents are often characterized by their extensive length; for instance, a typical 10-K filing can exceed 100,000 tokens and proprietary financial documents (like loan agreements or merger agreements) can run into more than 100 pages (Reddy et al., 2024). Recent studies have highlighted that LLM performance in Retrieval-Augmented Generation (RAG) and long-context Question Answering (QA) is sensitive not only to the sheer length of the context but also to the position of the relevant information within the context. LLMs often exhibit degraded performance when relevant facts are buried deep within lengthy contexts, compared to when those facts appear at the beginning or end, known as the 'lost in the middle' problem (Liu et al., 2024). Existing hallucination benchmarks mostly focus on short-context scenarios, failing to address the challenges LLMs face when processing the lengthy, dense documents typical in finance, such as SEC filings (Lin et al., 2021). Critically, methods used to create existing long-context benchmarks often rely on artificially concatenating unrelated documents, which fails to replicate the cohesive, topic-specific nature of real-world financial long-context QA tasks, potentially leading to misleading evaluation results.

To address these gaps, we introduce PHANTOM, a first of its kind benchmark dataset specifically designed for evaluating hallucination detection methods in long-context QA within the finance domain, using SEC filings as the primary data source. Our core contribution lies not only in the dataset itself but also in the methodology used for its creation. We begin with generating a high-quality 'seed' dataset by first extracting random 500-token chunks ('seed' chunks) from diverse Securities and Exchange Commission (SEC) filings and then using these as context to generate high-quality "query-answer-document (chunk)" triplets, with either hallucinated or correct answers. The quality of these triplets is partially validated by human annotators. A primary innovation of our work lies in the systematic expansion of this seed dataset to create long context datasets with various context length and information placement. Leveraging the validated seed dataset, we create extended context versions with lengths of 2000, 5000, 10000, 20000, and 30000 tokens. For each context length, we generate three distinct variations: one where the original 500-token seed chunk (containing the information needed to answer the query) is placed at the beginning, one where it is placed in the middle, and one where it is placed at the end of the extended context. This extension is achieved by incorporating contiguous text from the original source SEC filing, preceding and/or succeeding the seed chunk as required. This generation strategy provides several key advantages:

1. **Efficiency and Validity Preservation**: By reusing the validated queries and answers from the seed dataset, we circumvent the need for additional costly human validation for the long-context variations. The fundamental relationship between the query, the answer (both correct and hallucinated), and the necessary supporting text within the seed chunk remains constant.

2. **Contextual Realism**: Unlike synthetic approaches that use synthetic documents or concatenate unrelated documents to achieve length, our method extends context within the original SEC filing document. This ensures that the extended context maintains the natural flow, topic coherence, and stylistic properties of real-world financial documents, presenting a more realistic challenge for hallucination detection methods.

3. **Controlled Experimental Design**: Since the query, answer and label remain identical across all derived samples (different lengths and positions) originating from the same seed sample, our dataset enables fair and direct comparisons of model performance sensitivity to context length and information placement.

PHANTOM provides a valuable resource for the research community to benchmark and improve hallucination detection methods in the financial domain, particularly focusing on the challenges posed by reasoning and information retrieval in emerging AI application domains, where hallucinations can have significant impact on trust and reputation. Our dataset facilitates a detailed assessment of hallucination detection methods across different context lengths and allows us to rigorously evaluate the efficacy of attention mechanisms in mitigating information loss. This paper details the dataset construction methodology and presents benchmark results of state-of-the-art open-weight and closed-source LLMs on our datasets. This evaluation results have provided valuable insights into models' hallucination detection capabilities and how well models maintain performance over different context sizes and locations of relevant information. These comprehensive insights not only highlight current weaknesses in models, but also demonstrate how the proposed datasets can be used to fine-tune LLMs to address these weaknesses and lead to more efficient detection strategies.

## 2 Related Work

Our work builds upon several existing efforts in evaluating systems using synthetic benchmarks. HaluEval (Li et al., 2023) introduces a synthetically generated and human-annotated hallucination evaluation benchmark covering QA, dialogue and summarization but focuses on short context and general domain. HaluEval 2.0 (Li et al., 2024a) extended HaluEval with experiments that probe the sources and mitigation of hallucination. It covers the financial domain but focuses on opinionated QA without context. ARES (Saad-Falcon et al., 2024) fine-tunes judges to benchmark RAG systems along the dimensions of context relevance, answer faithfulness, and answer relevance. In contrast to our work, it requires an -albeit small- handcrafted dataset of human preference labels and is not situated within the finance domain. In comparison, our work does not require human labeling effort, which can be quite difficult to obtain in specialized domains, such as finance. RAGEval (Zhu et al., 2024) leverages synthetic data generation for domain-specific datasets to assess RAG systems. While the work assesses systems in finance, among others, its hallucination detection discussion is single-focused on finding contradictory facts in an answer and lacks detail. This contrasts with our benchmark, which offers a more nuanced and comprehensive evaluation of hallucination across longer contexts and a broader range of error types.

AlphaFin (Li et al., 2024b) and FinMTEB (Tang and Yang, 2025) contribute important financial benchmarks; however, they predominantly emphasize traditional financial analysis and retrieval tasks and do not specifically investigate hallucination detection. Similarly, OmniEval (Wang et al., 2024) has provided an synthetic framework/dataset for evaluating RAG systems in the financial domain. Despite its comprehensive evaluation of RAG systems along multiple domains (including hallucination), OmniEval does not consider the significant performance degradation over extended context lengths. This is a critical shortcoming given the growing use of generative language models where context size can heavily impact the model's behavior.

Additionally, Hallusionbench (Guan et al., 2024) offers a sophisticated diagnostic suite for identifying hallucinations in vision-language models. While Hallusionbench effectively dissects visual and textual hallucination phenomena, its focus lies outside the financial QA paradigm and extended context evaluation that our work addresses. Finally, another recent work presents a dataset and method for detecting hallucinations in long contexts (Liu et al., 2025). However, their work is less relevant to the financial domain due to two key limitations. First, their dataset does not incorporate the specific linguistic and logical complexities inherent in financial documents which has a significantly impact on the types of hallucinations that could occur. Second, the average context length in their dataset (5,101 tokens) is considerably shorter than the extended context lengths (up to 30,000 tokens) offered by PHANTOM, limiting its ability to assess the impact of very long financial documents on hallucination detection performance.

We show that by benchmarking state-of-the-art open weight generative LLMs, our dataset can provide valuable insights into how well a models can detect hallucinations, performance over varying context sizes, and the effectiveness of attention mechanisms. This integrated approach distinguishes our work as it encompasses broader evaluation metrics tailored for high-stakes financial decision-making that have not been simultaneously explored in previous studies. To the best of our knowledge, we are the first to present a dataset that combines all three areas of interest: hallucination detection, long context and the financial domain.

## 3 Data Creation

Before detailing the generation process, it is important to clarify the specific type of hallucination targeted by PHANTOM. Our focus is exclusively on source faithfulness. This concerns the model's ability to generate answers that are directly supported by and consistent with provided context. We do not include extrinsic hallucination, which relates to whether an answer aligns with real-world facts or external knowledge sources (Ji et al., 2023). In the context of financial open-book QA, the source faithfulness of LLM answers is critical for several reasons. First, financial decisions often rely on precise information extracted from specific documents or reports where linking conclusions directly to the source material is important. Second, the financial domain is characterized by rapid changes and nuanced interpretations, making it difficult to definitively assess the factuality of an answer based on external knowledge alone. Focusing on source faithfulness allows us to create a more controlled and reliable benchmark for evaluating LLMs in this critical area. Therefore, to avoid

creating ambiguity, an answer is considered "not hallucination" if it is faithful to the given context, and "hallucination" if it contradicts, misrepresents, or cannot be verified from the provided text, irrespective of its external factuality. This focus directly informs our data generation strategy, which centers on deriving both faithful and non-faithful answers relative to specific source context.

We selected the Securities and Exchange Commission (SEC) filings as our data source due to their public availability, authoritative nature, and representativeness of real-world financial communication. These documents offer diverse content and utilize nuanced, domain-specific language, providing a rich and challenging foundation for evaluating hallucination detection methods. Specifically, we sourced filings from the SEC's EDGAR database, focusing on four of the most common filing types to capture a range of financial reporting styles:

- **Form 10-K (Annual Report)**: Comprehensive yearly overview (business, financial, risks).
- **Form 8-K (Current Report)**: Time-sensitive reports on major corporate events.
- **Form 497K (Mutual Fund Summary Prospectus)**: Key fund information for investors (objectives, risks, fees).
- **Form DEF 14A (Proxy Statement)**: Information for shareholder voting (governance, compensation).

This selection provides variation in structure, purpose, and linguistic complexity, ensuring PHANTOM reflects realistic financial text diversity.

## 3.1 Seed Dataset Generation

The foundation of our benchmark is a seed dataset characterized by relatively short, focused contexts - seed chunks of 500 tokens. To be more specific, we call each sample in our seed dataset a seed sample, which contains a query, a 500-token seed chunk, an answer and a label. Our method provides a practical approach to generating synthetic financial open-book question-answering datasets containing subtle, yet realistic hallucinations. We used a Large language model (LLM) - Llama-3.3-70B-instruct (Dubey et al., 2024) to generate the seed dataset, which then acts as a building block for creating long-context datasets. The generation process of these seed samples is as follows:

1. **Seed chunk extraction**: From the SEC filing documents, we randomly extracted contiguous text chunks as our seed chunks. Each seed chunk was targeted to be approximately 500 tokens in length, using spaCy tokenizer (Honnibal and Montani, 2017). This length was chosen to be substantial enough to contain meaningful information for question answering but concise enough for efficient initial generation and validation.

2. **Query generation**: For each 500-token context chunk $d_i$, we employed Llama-3.3-70B-instruct, to generate a relevant question $q_i$ whose answer could be directly and fully inferred from $d_i$.

3. **Faithful answer generation**. Given a query $q_i$, along with the corresponding context (document chunk) $d_i$, we instructed the LLM to provide a accurate, source faithful answer $a_i$ to $q_i$ based only on $d_i$.

4. **Hallucinated answer generation**: Subsequently, for each sample $x_i = (q_i, d_i, a_i)$, we prompted the same LLM to generate another answer $a_i^*$ that might seem reasonable given the general topic but contradicts, misrepresents, or cannot be supported by $d_i$. This provides a new synthetic sample $x_i^* = (q_i, d_i, a_i^*)$ with hallucination. During this step, the LLM receives explicit instructions (with supporting examples[2] taken from real-world hallucinations) to deliberately craft answers containing subtle and difficult-to-detect hallucinations. This aims to mimic realistic LLM failure modes where the model might draw on its own knowledge incorrectly or confabulate details.

5. **Human validation**: To ensure the reliability of the generated seed samples, a manual verification process was employed. A subset ($38.4\%$) of the generated samples was independently inspected and validated by four domain experts. The validation focused on confirming the faithful answers were accurately and fully supported by the context chunk, and that the hallucinated answers were indeed incorrect or unverifiable based solely on the context chunk. Among the samples manually validated, $98.6\%$ of the samples were confirmed to be correct. This high accuracy is likely attributable to the short context length (500 tokens) used for seed dataset generation,

---

[2]Details of examples needed to be omitted as they contain private client information.

which falls well within the capabilities of state-of-the-art LLMs like Llama-3.3-70B-instruct. Furthermore, almost all validated hallucinated answers were confirmed to contain information that contradicted or was absent from the context chunk. This is not surprising given that the nature of our approach for generating the hallucinations - we specifically instruct the LLM to write answers that contain details that do not match or do not exist in the context chunk.

6. **Label Assignment**: Each validated sample yields two distinct data points for the benchmark:

   - (Query $q_i$, Context with length 500 token $d_i$, Ground Truth Answer $a_i$, Label = "Not Hallucination")
   - (Query $q_i$, Context with length 500 token $d_i$, Hallucinated Answer $a_i^*$, Label = "Hallucination")

This process resulted in 3962 seed samples with 1981 unique queries across 4 filing types as data source. The prompts used in the data generation process are shown in Appendix A.

## 3.2 Long-Context Dataset Generation

A primary contribution of our work is the systematic generation of long-context variations to study potential performance degradation of hallucination detection methods related to context length and information placement. This was achieved by extending the validated 500-token seed chunks using the original source SEC filing documents. Instead of artificially concatenating unrelated documents, our approach incorporates additional context around the seed chunk directly from the original filing document, ensuring that the resulting contexts remain realistic and coherent.

1. **Context expansion**: For each seed sample (Query, Context with length 500 token, Ground Truth Answer, Label), we identified the location of the 500-token chunk within its original, full SEC filing. We then programmatically extracted contiguous text preceding and/or succeeding this seed chunk from the source document to create longer contexts of target lengths: 2000, 5000, 10000, 20000 and 30000 tokens (approximated using the same tokenizer). Limiting the context length to 30,000 tokens strikes a balance between capturing the challenges of long-context processing in financial documents and maintaining computational feasibility for model evaluation; furthermore, 30,000 tokens is sufficient to observe the performance decrease of hallucination detection methods on long documents as shown in Section 4.

2. **Positional variation**: For each target length ($L = 2000, 5000, 10000, 20000, 30000$), we generated three versions by manipulating the position of the original 500-token seed chunk (which contains the necessary information to answer the query) within the extended $L$-token context:

   - Beginning: The 500-token seed chunk constitutes the first 500 tokens of the $L$-token context. The remaining $L - 500$ tokens are contiguous text immediately following the seed chunk in the original document.
   - Middle: The 500-token seed chunk is centered within the $L$-token context. Approximately $(L - 500)/2$ tokens preceding the chunk and $(L - 500)/2$ tokens succeeding the chunk are extracted from the original document to form the $L$-token context.
   - End: The 500-token seed chunk constitutes the final 500 tokens of the $L$-token context. The preceding $L - 500$ tokens are contiguous text immediately before the seed chunk in the original document.

3. **Query and answer preservation**: Crucially, the query, answer (both ground truth and hallucinated), and label associated with the seed sample are directly reused for all its derived long-context variations. The validity of the query-answer relationship is preserved because the necessary information (or lack thereof for hallucinations relative to the seed chunk) remains embedded within the extended context, specifically within that preserved 500-token seed chunk.

This novel structured approach ensures that comparisons across different context lengths and information positions are conducted using the same queries and answer candidates, providing a controlled environment that allows for fair, apples-to-apples comparisons across various long-context retrieval scenarios. Furthermore, by extending contexts with text from the original documents, we maintain a high degree of ecological validity compared to methods involving document concatenation. The reuse of validated seed samples significantly reduces the manual effort typically required for creating the large-scale, long-context benchmarks.

With the data generation methodology established, the subsequent sections transition towards understanding and utilizing the resulting benchmark. In Section 4.1, we delve into a content analysis of PHANTOM, examining the financial topics covered and characterizing the type of hallucination included in the dataset. Following this analysis, Section 4 demonstrates the practical application of the dataset by presenting experimental results from benchmarking various hallucination detection methods, and characterizing their robustness to increasing context lengths and varying information placement.

# 4 Experiments

## 4.1 Content Analysis

Before delving into the experimental results we provide a fine-grained analysis of the PHANTOM dataset, which we created using the methodology explained in the previous section. Our analysis explores two themes, we first investigate what types of hallucinations are present in our dataset, followed by the financial themes that the dataset covers. For an overview of our results refer to Figure 1. For the purpose of reproducibility, we provide the details of prompts and methods in Appendices A and C.

In line with our definition of hallucination as mentioned in Section 3, we identify three major classes of hallucination types: Class 1 - The answer to the question contradicts the provided context. Class 2 - The answer to the question misrepresents the provided context. Class 3 - The answer to the question cannot be verified from the provided context, irrespective of its external factuality.

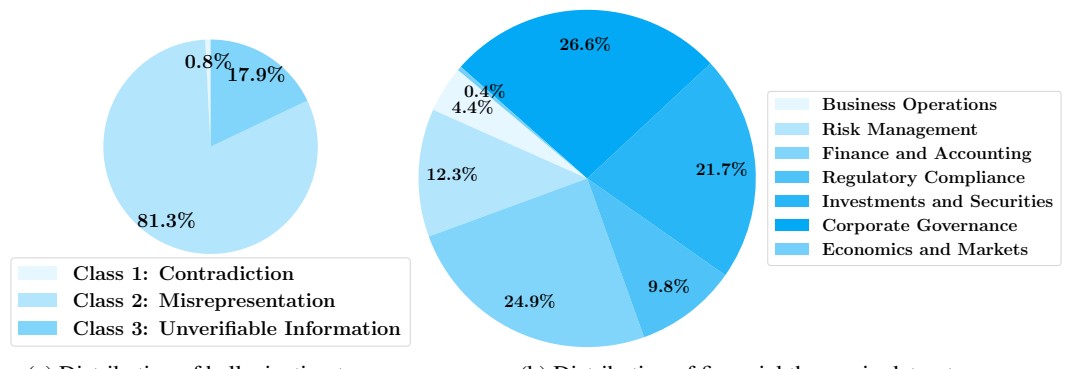

(a) Distribution of hallucination types.

(b) Distribution of financial themes in dataset.

Figure 1: Dataset content analysis.

For the purpose of classification we incorporate these definitions into an LLM prompt, which is shown in Listing 1. Figure 1a shows the results on the dataset combining all filing types. We find that the majority of the hallucinations (81.3%) are cases where the answer misrepresents the provided context. This is expected as the model is instructed to create hallucinations that are "minor", "subtle" and "hard to catch", according to our dataset generation prompt (Listing 6). Surprisingly, only a very small amount (0.8%) of instances are classified as contradicting the context. The reason for this might also stem from the fact that contradictions are less "subtle" or "hard to catch". The final class of hallucinations are questions that cannot be verified from the provided context (17.9%), which are cases where the model makes up facts and scenarios.

Figure 1b presents the distribution of financial themes across the dataset. We find that the most prominent categories are distributed among different filing types, e.g., "Finance and Accounting" (24.9%) from 10-K filings, "Investments and Securities" (21.7%) from 8-K filings, "Corporate Governance" (26.6%) from DEF 14A filings. Taking this into consideration, we argue that the dataset exhibits a mixture of queries cutting across various topics of interest to financial users/applications, which points to the utility of using different filing types in the dataset augmentation process.

## 4.2 Evaluating Hallucination Detection Models

In our experimental evaluation, we aim to show the utility of PHANTOM to investigate important research questions and benchmark hallucination detection abilities of various open-weight and closed-source LLMs. All experiments were conducted on standardized hardware with consistent hyper-parameter settings (Appendix D) and prompts (Appendix A) to ensure fair comparisons across models.

We aim to answer the following research questions: RQ1) How do state-of-the-art open-weight models compare against leading closed-source LLMs? RQ2) What are the performance trade-offs when using smaller open-weight models with varying parameter sizes? RQ3) How do reasoning-focused distillation or fine-tuning impact performance?

To answer these questions, we evaluate a diverse set of models. For open-weight models, we test Llama-3.3-70B-Instruct (Dubey et al., 2024), the Qwen series (2.5 and 3) (Yang et al., 2024, 2025), Microsoft's Phi-4 (Abdin et al., 2024), and distilled models from the DeepSeek-R1-Distill series (Guo et al., 2025). To benchmark against proprietary systems (RQ1), we include leading closed-source models: Google's Gemini series (Gemini Team et al., 2023, 2024; Google, 2024), and OpenAI's GPT-4o (Hurst et al., 2024) and o3-mini (OpenAI, 2025). The inclusion of different Qwen model sizes allows us to analyze performance trade-offs (RQ2), while the DeepSeek-R1-Distill models and the comparison between Qwen3's 'Instruct' and 'Thinking' variants help us address RQ3.

Table 1 summarizes key performance metrics for all models on the seed dataset with 500 token context size. Regarding RQ1, while Llama-3.3-70B-Instruct achieves the highest F1 score (0.916), this result should be interpreted with caution. As this model was used to generate the seed dataset, which may inflate its performance. More revealingly, top-performing medium-sized open-weight models like Qwen3-30B-A3B-Thinking and Phi-4 outperform all tested closed-source models. This leads to RQ2, we find a large performance drop for smaller models. Especially, the 7B variant of Qwen-2.5 performs only slightly above random guess. Finally, for RQ3, the 'Thinking' variant of Qwen3 outperforms its 'Instruct' counterpart, boosting the F1 score from 0.848 to 0.882, showing the benefit of reasoning-focusing tuning. In contrast, the DeepSeek distillation models present a trade off, they tend to show improvements on precision, but at a large cost of recall compared to their base models.

Table 1: Hallucination detection performance of various open-weight and closed-source LLMs on PHANTOM seed dataset.

| Model | Accuracy | Precision | Recall | F1 Score |
|---|---|---|---|---|
| Llama-3.3-70B-Instruct | 0.916 | 0.920 | 0.912 | 0.916 |
| Qwen2.5-7B-Instruct | 0.589 | 0.549 | 0.998 | 0.708 |
| Qwen2.5-14B-Instruct | 0.862 | 0.816 | 0.938 | 0.872 |
| Qwen2.5-32B-Instruct | 0.829 | 0.766 | 0.954 | 0.849 |
| Qwen3-30B-A3B-Thinking | 0.870 | 0.818 | 0.957 | 0.882 |
| Qwen3-30B-A3B-Instruct | 0.823 | 0.747 | 0.983 | 0.848 |
| Phi-4 | 0.874 | 0.816 | 0.967 | 0.885 |
| DeepSeek-R1-Distill-Qwen-7B | 0.632 | 0.667 | 0.530 | 0.590 |
| DeepSeek-R1-Distill-Qwen-14B | 0.864 | 0.895 | 0.824 | 0.858 |
| DeepSeek-R1-Distill-Qwen-32B | 0.874 | 0.893 | 0.849 | 0.870 |
| DeepSeek-R1-Distill-Llama-70B | 0.867 | 0.926 | 0.799 | 0.858 |
| Gemini-1.5-flash-002 | 0.855 | 0.804 | 0.940 | 0.867 |
| Gemini-2.0-flash | 0.882 | 0.938 | 0.818 | 0.874 |
| o3-mini | 0.838 | 0.925 | 0.734 | 0.818 |
| GPT-4o | 0.867 | 0.868 | 0.866 | 0.867 |

PHANTOM also enables us to investigate how models perform when given long contexts, a common scenario in financial question answering. Figure 2 and Figure 3 present the Precision and Recall for various open-weight and closed-source models, respectively, across different context lengths (2000, 5000, 10000, 20000, and 30000 tokens) and varying positions of the relevant information ('Beginning', 'Middle', 'End'). Performance is averaged over 10-K and DEF 14A filing types, as 8-Ks and 497Ks are not long enough to generate contexts exceeding 20,000 tokens. We feature the

most relevant models in the main text, and for completeness, results for all models are provided in Appendixe E (Figure 4, 5, 6 and 7).

A discernible trend across nearly all models is the degradation of performance as the context length increases, highlighting the inherent challenge of detecting hallucinations in extensive financial documents. This trend, however, manifests differently between open-weight and closed-source models. For the open-weight LLMs (Figure 2), this performance degradation is particularly severe. We observe a sharp decline in performance as context length grows, with a near-total collapse in recall for contexts larger than 20,000 tokens. In contrast, the closed-source models (Figure 3) demonstrate greater resilience to long contexts. While they also experience a performance drop with increasing context length, the degradation is far more graceful. This suggests that while nearly no model is immune to the challenges of long-context reasoning, leading closed-source models provide a substantial advantage in robustness. We believe that precisely quantifying this performance disparity is a key contribution of this work. It not only highlights a significant weakness in current open-source models but also underscores the utility of PHANTOM as a crucial tool for diagnosing such failures and benchmarking progress in long-context detection tasks.

Further, we see evidence that the placement of relevant information also affects performance. Generally, models exhibit higher precision when the relevant information is located at the 'Beginning' of the context. In terms of recall the pictures is less clear as there is no obvious difference between 'Beginning', 'Middle' or 'End'. We also find that distilling R1 into the Llama model has benefits for hallucination detection in large contexts, as it improves both precision and recall up to 20,000 tokens.

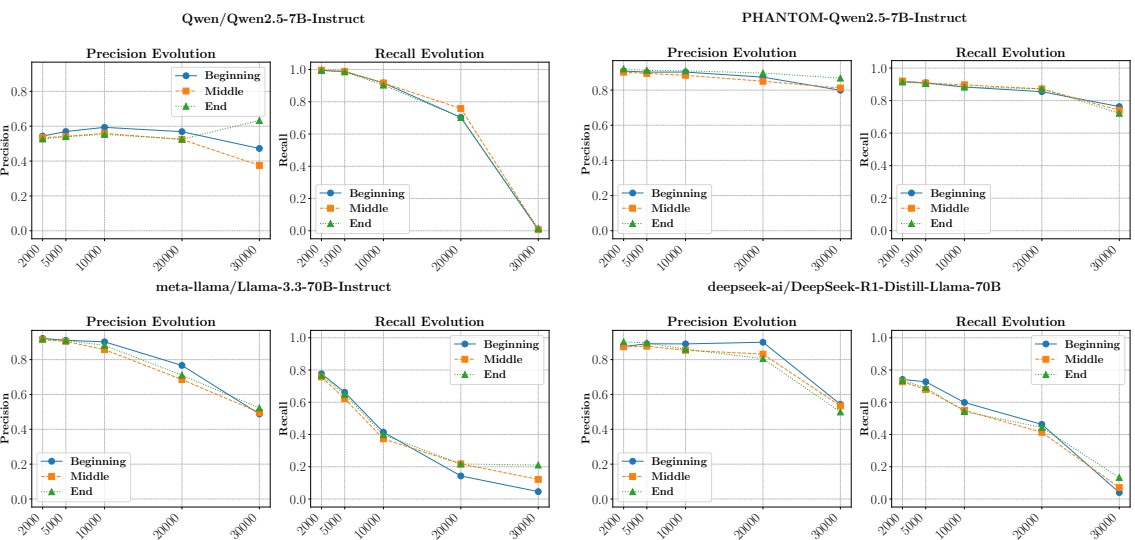

Figure 2: Hallucination detection performance for long contexts. X-axis is the number of tokens.

# 5 Fine-tuning a Hallucination Detection Model

## 5.1 Model Training

To showcase how PHANTOM can be used to enhance the capabilities of hallucination detection, we performed supervised fine-tuning (SFT) on the Qwen2.5-7B-Instruct model. The training data was generated using the Llama-3.3-70B-instruct model on Form 8-K and Form 497K seed datasets. Specifically, we prompted a Llama-3.3-70B-instruct model to generate reasoning chains and hallucination labels for each 8-K and 497K seed sample. Since our dataset contains ground truth labels, we employed rejection sampling, discarding any training data points where the generated label did not align with the ground truth. The Qwen model was then fine-tuned using the SFTTrainer from the trl (Transformer Reinforcement Learning) package (von Werra et al., 2020) on a next-token prediction objective, training the model to learn the reasoning patterns and accurately predict the hallucination label based on query, context and answer. The model was trained for 3 epochs with a sequence length of 4,096 tokens. We utilized Parameter-Efficient Fine-Tuning (PEFT) with LoRA (Low-Rank

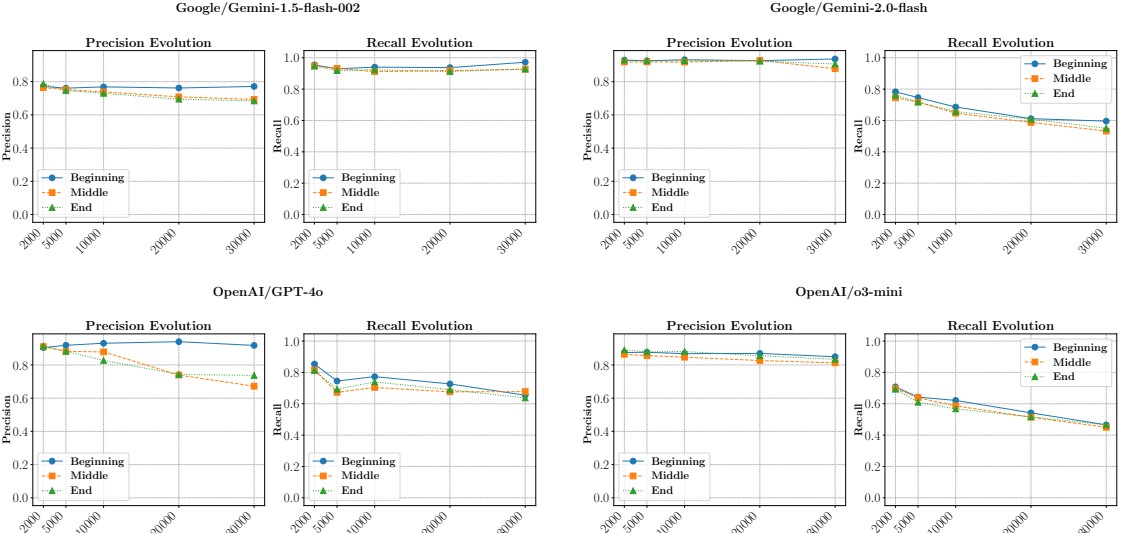

Figure 3: Hallucination detection performance of closed-source models for long contexts. X-axis is the number of tokens.

Adaptation) (Hu et al., 2022) to reduce computational costs and memory footprint during training, while maintaining performance.

## 5.2 Model Evaluation

To assess the effectiveness of fine-tuning on PHANTOM and especially, to measure the fine-tuned model's ability to generalize to entirely unseen document types, the fine-tuned model, which we call PHANTOM-Qwen2.5-7B-Instruct, is evaluated on the 10-K and DEF 14A datasets, while the training process exclusively used data from 8-K and 497K filings. We note that our dataset's structure also allows for in-distribution evaluations, should researchers wish to split data within a single filing type for training and testing.

We find that the model improves in terms of F1-score compared to the Qwen2.5-7B-Instruct base model from 0.705 to 0.933 and 0.706 to 0.923 for 10-K and DEF 14 A, respectively. Compared to our strongest model thus far, which is Llama-3.3-70B-Instruct, the finetuned model performs about equally on 10-K, 0.932 versus 0.933; and improves on DEF 14A, 0.880 versus 0.923. We show the full performance results comparing to all models in Table 7 in the appendix.

We are specifically interested in evaluating PHANTOM-Qwen2.5-7B-Instruct in long context scenarios. The top right plots of Figure 2 allow us to compare the finetuned model's performance to its base model and the two Llama 3.3 variants. We find PHANTOM-Qwen2.5-7B-Instruct to improve over its base model on almost all data points on precision and recall. Substantial performance improvements are observed especially for context sizes of 20k and above, even compared with most of the closed-source models tested (Figure 3). We believe this to be promising evidence that finetuning on a domain-specific dataset can help improve a model's performance on long contexts as we find PHANTOM-Qwen2.5-7B-Instruct to improve over the base model. We realize that these impressive improvement ought to be validated on real-world hallucination examples from out of sample datasets. We plan to do further investigations to better understand the model's performance in this regard. To shed further light on these improvements we conducted a qualitative analysis of the reasoning traces of the base model and our fine-tuned version. We find that the improvement in recall for larger contexts is mostly due to the base model missing out on nuanced hallucinations. We provide some specific examples in Appendix F.

## 6 Limitations

While PHANTOM introduces a valuable benchmark for hallucination detection in financial long-context QA, it's important to acknowledge several limitations. Firstly, the dataset is generated using

LLMs, specifically Llama-3.3-70B-instruct, for query, faithful answer, and hallucinated answer creation. Although human validation is performed on a subset of the seed dataset, the synthetic nature of the data generation process might introduce biases or patterns that do not fully reflect real-world hallucination scenarios in financial documents. The types of hallucinations generated are also influenced by the specific prompts and instructions given to the LLM, potentially limiting the diversity of hallucination types captured in the dataset. In Section 4.1 we performed a thorough analysis of hallucination types in the dataset, but as the theoretical number of types of hallucinations is unlimited, it is possible that additional types exist that were not captured by our analysis. Secondly, the context expansion method, while preserving topic coherence, relies on contiguous text from SEC filings. This approach may not fully capture the complexities of information retrieval in scenarios where relevant information is scattered across non-contiguous sections of a document or across multiple documents in a multi-document RAG setting. Thus, when context is expanded, more relevant information can be introduced that potentially can alter the resulting answer to a given question. Furthermore, the dataset focuses exclusively on source faithfulness, not considering extrinsic hallucinations that relate to alignment of answers with real-world facts or external knowledge sources, which are out of scope of this work.

Finally, the benchmark primarily utilizes SEC filings, which, while being representative of real-world financial documents, may not encompass the full spectrum of financial document types. Expanding the dataset to include other document types, such as earnings call transcripts, loan agreements, and merger agreements, could enhance its generalizability. Further, many financial documents incorporate multimodal elements like charts, graphs, and images, which provide crucial information. Future iterations of the benchmark should consider incorporating documents with multimodal contexts to better reflect the complexities of real-world financial analysis.

# 7   Conclusion and Future Work

Mitigating the risks associated with LLM hallucination is crucial for deploying AI reliably in the financial sector. In this paper, we introduce PHANTOM, a novel benchmark dataset specifically constructed to evaluate capabilities of hallucination detection methods for financial long context QA tasks. Our key contribution is the generation of a validated seed dataset along with systematic variations across extensive context lengths (up to 30,000 tokens) and different placements of relevant information. By leveraging contiguous text from original documents for realism and reusing validated query-answer pairs, we enable controlled assessment of how context manipulation impacts the difficulty of detecting hallucinations. Our benchmarking experiments (Section 4) demonstrate that context length and information placement significantly affect the performance of current hallucination detection strategies, highlighting the utility of PHANTOM for rigorously assessing these critical capabilities. These findings suggest that current methods struggle with long contexts and varying information placement. The observed performance trends revealed by PHANTOM have critical implications for deploying LLMs in financial applications. Specifically, the sensitivity to context length and information placement underscores the need for careful design of RAG systems in finance. Strategies such as context re-ranking, summarization, and improved attention mechanisms are crucial to mitigate the risk of hallucination and ensure the reliability of AI-driven financial decision-making. Further research into these mitigation strategies, guided by benchmarks like PHANTOM, is essential for responsible AI adoption in the financial domain.

This work opens several avenues for future research. Firstly, the dataset could be expanded by incorporating a wider variety of financial document types (e.g. other SEC filing types, earnings call transcript, loan agreements etc.) and documents with multimodel elements like charts, graphs, and images. Secondly, PHANTOM serves as a crucial resource for developing and refining novel hallucination detection techniques, particularly those optimized for long-context financial text. Further research could involve analyzing why certain detection methods fail under specific long-context conditions, potentially leading to more robust detector designs informed by common LLM failure patterns in this domain. Ultimately, we hope PHANTOM will foster advancements in automated faithfulness verification, contributing to the development of more reliable and trustworthy AI systems for financial information processing.

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

# A Prompts

This section contains the prompts that were used in our work.

Listing 1: Dataset Analysis: Hallucination Type Classification.

```
Your task is to investigate the following QUESTION, ANSWER and CONTEXT. We know the
ANSWER to be hallucinated. Your task is to determine what kind of hallucination it
is.

QUESTION:
{question}

ANSWER:
{answer}

CONTEXT:
{context}

You will classify the type of hallucination in any of the following three classes:
Class 1: The ANSWER to the QUESTION contradicts the provided CONTEXT.
Class 2: The ANSWER to the QUESTION misrepresents the provided CONTEXT.
Class 3: The ANSWER to the QUESTION cannot be verified from the provided CONTEXT,
irrespective of its external factuality.

Please output the class number only, without any additional text. Also, you need
to choose only one class.
```

Listing 2: Dataset Analysis: Extract financial categories.

```
You are designated as an assistant that identify and extract high-level categories
from list of questions.
You should avoid giving specific details and provide unique categories solely.
The List of Questions is a python list seperated by comma below.

Your output should only contain the following details:
1. List of high level topics generates.
2. Table containing the topics and number of questions belonging to that category.

List of Questions: {questions}
```

Listing 3: Dataset Analysis: Classify financial questions

```
Your task is to classify the question from one of the below list of categories:
Note that some questions may belong to multiple categories but assign them to the
most relevant category based on the content of the question.

Categories:
{categories}

Question: {question}
Your output should only include the category of the question.
```

Listing 4: Query Generation Prompt.

```
You are a financial analyst. You are asked to write 1 question that can be answered
by the information in the provided document chunk which is from 10-K filings.
***[START OF DOCUMENT CHUNK]
{chunk}
***[END OF DOCUMENT CHUNK]

Guidelines:
1. You are a financial analyst. Imagine you're given a tool that you can ask
questions about a SEC filing so that it saves your time reading the document.
```

```
Come up with a question that you want to ask. It needs to be financial
meaningful.
2. Try to avoid asking questions about small unmeaningful details in the SEC
filings. Come up with realistic questions that a financial analyst might care
about.
3. Make sure the question you generated can be answered with the information in
the document chunk and does not require any other knowledge or information.

You must return ONLY the question. Do not generate anything else other than the
question.
```

Listing 5: Ground Truth Answer Generation Prompt.

```
Answer the following question based on the information in the given document
chunk.
[QUESTION]
{query}
[DOCUMENT CHUNK]
{chunk}

Provide answer to the QUESTION only using information from the DOCUMENT CHUNK
provided. Make sure your answer is consistent with the information in the
document chunk.
You answer should only include information that is supported by the document
chunk.
If the document chunk does not contain enough information for you to answer the
question, output "Information is not available in the document."
You must output ONLY the answer. Do not generate anything else other than the
answer.
```

Listing 6: Synthetic Hallucination Creation Prompt.

```
Given a question, a correct answer and a reference document chunk, write a
HALLUCINATION ANSWER to the question.

[QUESTION]
{query}
[A CORRECT ANSWER]
{answer}
[DOCUMENT CHUNK]
***
{chunk}
***

[INSTUCTION]
- You're given a CORRECT ANSWER to the QUESTION. The CORRECT ANSWER provided is
        consistent with the information in the DOCUMENT CHUNK.
- This HALLUCINATION ANSWER you need to write is mostly correct, but contains
        information that is not fully supported by the DOCUMENT CHUNK. The
        unsupported  content in the HALLUCINATION ANSWER is minor and subtle.
- Be creative with writing the HALLUCINATION ANSWER. You know the domain
        terminology and jargon very well. Make it realistic sounding and hard to
        catch even for a domain expert.
- For example, write an answer that is mostly correct, but contains a small detail
        that does not match the context in the document chunk, or one part of the
        answer talks about something that does not exist in the document chunk, or
        one part of the answer is missing some details that causes the answer to be
        misleading.

[EXAMPLES]
Here are some examples of HALLUCINATION ANSWERs that are hard to catch.

[EXAMPLE 1]
```

```
[EXAMPLE 2]

[EXAMPLE 3]

[END OF EXAMPLES]

You must output ONLY the HALLUCINATION ANSWER. Don't write anything other than the
HALLUCINATION ANSWER itself.
```

## B Dataset Statistics

Table 2: Number of samples for seed dataset (500 context size) by filing type

| Dataset | Number of samples | Positive Samples | Negative Samples |
|---|---|---|---|
| **10-K Filing** | 994 | 497 | 497 |
| **8-K Filing** | 986 | 493 | 493 |
| **497-K Filing** | 994 | 497 | 497 |
| **DEF-14A Filing** | 988 | 494 | 494 |

Table 3: Number of samples for long context dataset by filing type and position. Each dataset is balanced i.e. equal distribution of positive and negative samples

| Dataset | Position | 2k | 5k | 10k | 20k | 30k |
|---|---|---|---|---|---|---|
| **10-K Filing** | Beginning | 984 | 896 | 892 | 886 | 850 |
| | Middle | 980 | 904 | 896 | 882 | 846 |
| | End | 982 | 896 | 892 | 882 | 846 |
| **DEF-14A Filing** | Beginning | 1000 | 996 | 980 | 840 | 654 |
| | Middle | 1000 | 996 | 980 | 840 | 654 |
| | End | 1000 | 996 | 980 | 840 | 654 |

## C Complementary Information for Content Analysis

As large-scale analysis is prohibitively expensive due to the need for human labels, it is a common practice to utilize LLMs to aid in this task. Thus, our main content analysis relies on an LLM to classify each datapoint within the dataset according to its hallucination type or financial theme. For analyses we utilize the Llama 3.3 70B instruct (Dubey et al., 2024) model, which is in line with our model used for data generation.

We are especially interested in what types of questions are contained in the dataset in relation to the financial themes they exhibit. For this we employ a two-step approach to discover these financial themes: we first pass all questions in a dataset as a prompt to the model and ask it to come up with a list of financial themes. Appendix A Listing 2 shows the prompt that we employed in this process. After obtaining an initial list of themes, we manually prune the list by removing infrequent and redundant categories. In the second round we individually classify each question in the dataset into the selected categories. For this we again leveraged the LLM using the prompt presented in Appendix A Listing 3. We randomly reorder the list of categories before sending them to the prompt. Then, for each question, we execute three independent runs and use majority voting on the results to determine its final category. This approach aims to improve robustness of the classification outcome.

## D Hyperparameters

Table 4 lists the hyperparamters used in our finetuning experiment. Table 5 lists the hyperparamters used when benchmarking the models. All experiments were executed on a cloud instance with the properties listed in Table 6.

Table 4: Hyperparameters used for finetuning Qwen2.5-7B-Instruct

| Configuration | Hyperparameter | Value |
|---|---|---|
| **LoraConfig** | lora_alpha | 128 |
| | lora_dropout | 0 |
| | rank | 64 |
| | task_type | CAUSAL_LM |
| | bias | none |
| **SFTConfig** | max_length | 4096 |
| | num_train_epochs | 3 |
| | gradient_accumulation_steps | 4 |
| | learning_rate | 2e−4 |
| | optimizer | adamw_8bit |
| | weight_decay | 0.01 |
| | lr_scheduler_type | linear |

Table 5: Hyperparameters used for generating seed dataset and benchmarking PHANTOM dataset.

| Method | Hyperparameter | Value |
|---|---|---|
| **Seed Dataset Generation** | max_new_tokens | 5000 |
| | top_p | 0.9 |
| | top_k | 50 |
| **Benchmarking PHANTOM Dataset** | max_new_tokens | 8000 |
| | temperature | 0.7 |
| | top_p | 0.8 |
| | repetition_penalty | 1.05 |

# E   Complimentary Experimental Results

Table 7 lists the F1-score performance for each LLM on the PHANTOM dataset. The table lists aggregate as well as individual performance scores over different filing type subsets. We list results for our finetuned model (PHANTOM-Qwen2.5-7B-Instruct) for 10-K and DEF 14A only, as it was trained on 8-K and 497-K datasets.

We also present the complete long-context experiment results for all evaluated models. Due to the large number of models, the results are organized into four figures for clarity.

Figures 4 and 6 detail the performance of several open-weight models alongside their distilled counterparts. This allows for a direct comparison of how knowledge distillation affects performance on our benchmark across different models.

Figures 5 and 7 presents the results for a broader range of models, including closed-source models, other state-of-the-art open-weight models and our fine-tuned model. For Phi-4, since it only supports context window up to 16k tokens, we are not able to test it on datasets with 20k tokens and above.

# F   Example for Qwen2.5-7B-Instruct and PHANTOM-Qwen2.5-7B-Instruct

Figure 8 provides example outputs of Qwen2.5-7B-Instruct and PHANTOM-Qwen2.5-7B-Instruct for comparison.

Table 6: Details on compute instance used for experiments.

| Property | Value |
|---|---|
| GPU Count | 8 GPUs |
| GPU Memory | 640 GB |
| vCPU Count | 208 vCPUs |
| VM Memory | 1,872 GB |

Table 7: Hallucination detection F1 score of various open-weight and closed-source LLMs on PHANTOM seed dataset for each filing type

| Model | 10-K | 8-K | 497-K | DEF 14A | Mean F1 Score |
|---|---|---|---|---|---|
| Llama-3.3-70B-Instruct | 0.932 | 0.933 | 0.917 | 0.880 | 0.916 |
| Qwen2.5-7B-Instruct | 0.705 | 0.708 | 0.714 | 0.706 | 0.708 |
| Qwen2.5-14B-Instruct | 0.897 | 0.896 | 0.868 | 0.828 | 0.872 |
| Qwen2.5-32B-Instruct | 0.865 | 0.888 | 0.828 | 0.815 | 0.849 |
| Qwen3-30B-A3B-Thinking | 0.909 | 0.920 | 0.861 | 0.836 | 0.882 |
| Qwen3-30B-A3B-Instruct | 0.881 | 0.865 | 0.837 | 0.811 | 0.848 |
| Phi-4 | 0.900 | 0.924 | 0.865 | 0.851 | 0.885 |
| DeepSeek-R1-Distill-Qwen-7B | 0.627 | 0.613 | 0.550 | 0.568 | 0.590 |
| DeepSeek-R1-Distill-Qwen-14B | 0.891 | 0.872 | 0.867 | 0.800 | 0.858 |
| DeepSeek-R1-Distill-Qwen-32B | 0.885 | 0.891 | 0.865 | 0.839 | 0.870 |
| DeepSeek-R1-Distill-Llama-70B | 0.869 | 0.863 | 0.874 | 0.824 | 0.858 |
| Gemini-1.5-flash-002 | 0.890 | 0.874 | 0.853 | 0.849 | 0.867 |
| Gemini-2.0-flash | 0.891 | 0.876 | 0.875 | 0.854 | 0.874 |
| o3-mini | 0.853 | 0.840 | 0.811 | 0.768 | 0.818 |
| GPT-4o | 0.914 | 0.895 | 0.855 | 0.804 | 0.867 |
| PHANTOM-Qwen2.5-7B-Instruct | 0.933 | N/A | N/A | 0.923 | N/A |

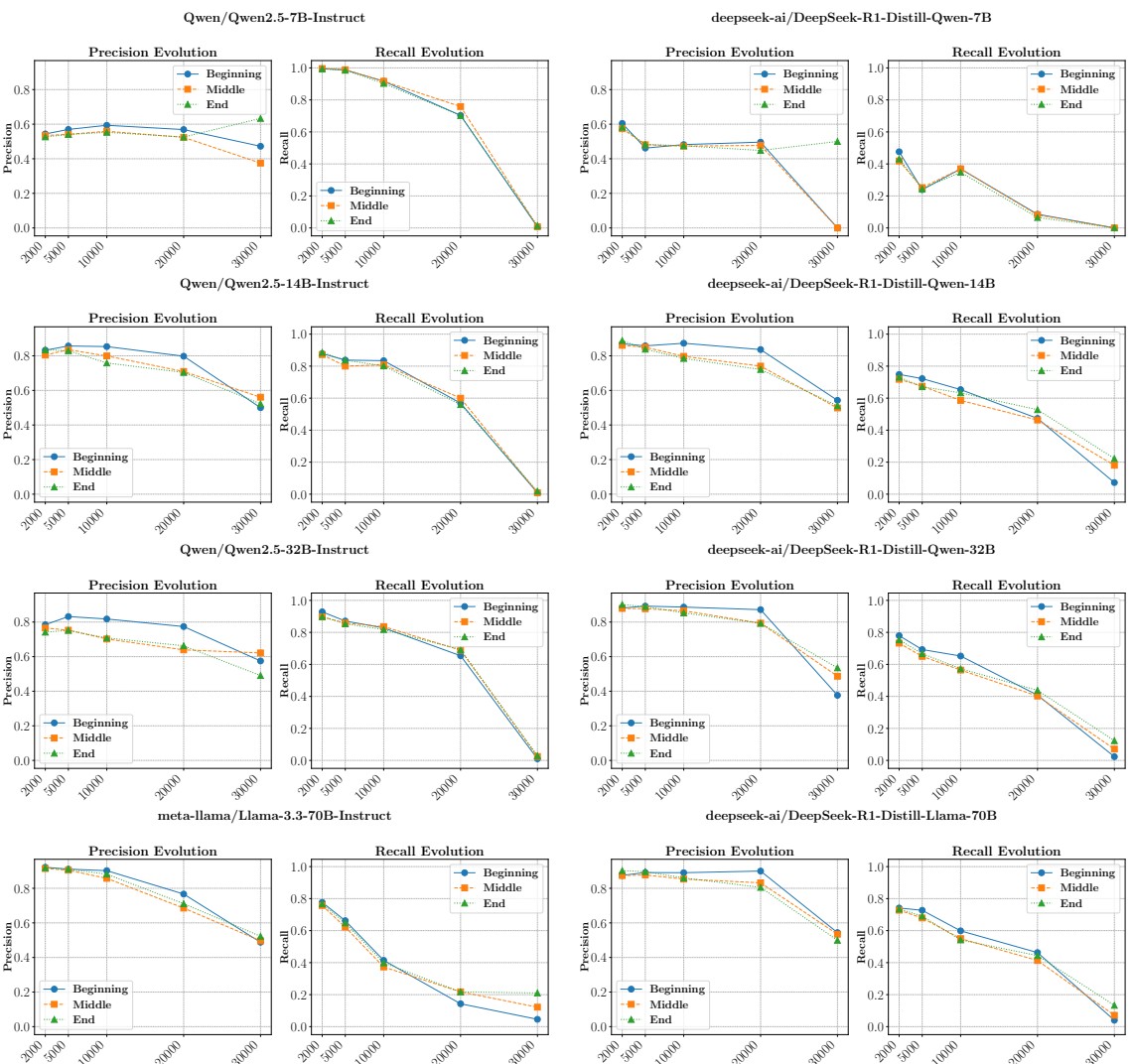

Figure 4: Precision and recall on the long-context benchmark for base models (left column) and their corresponding distilled versions (right column). The x-axis represents the context length in tokens.

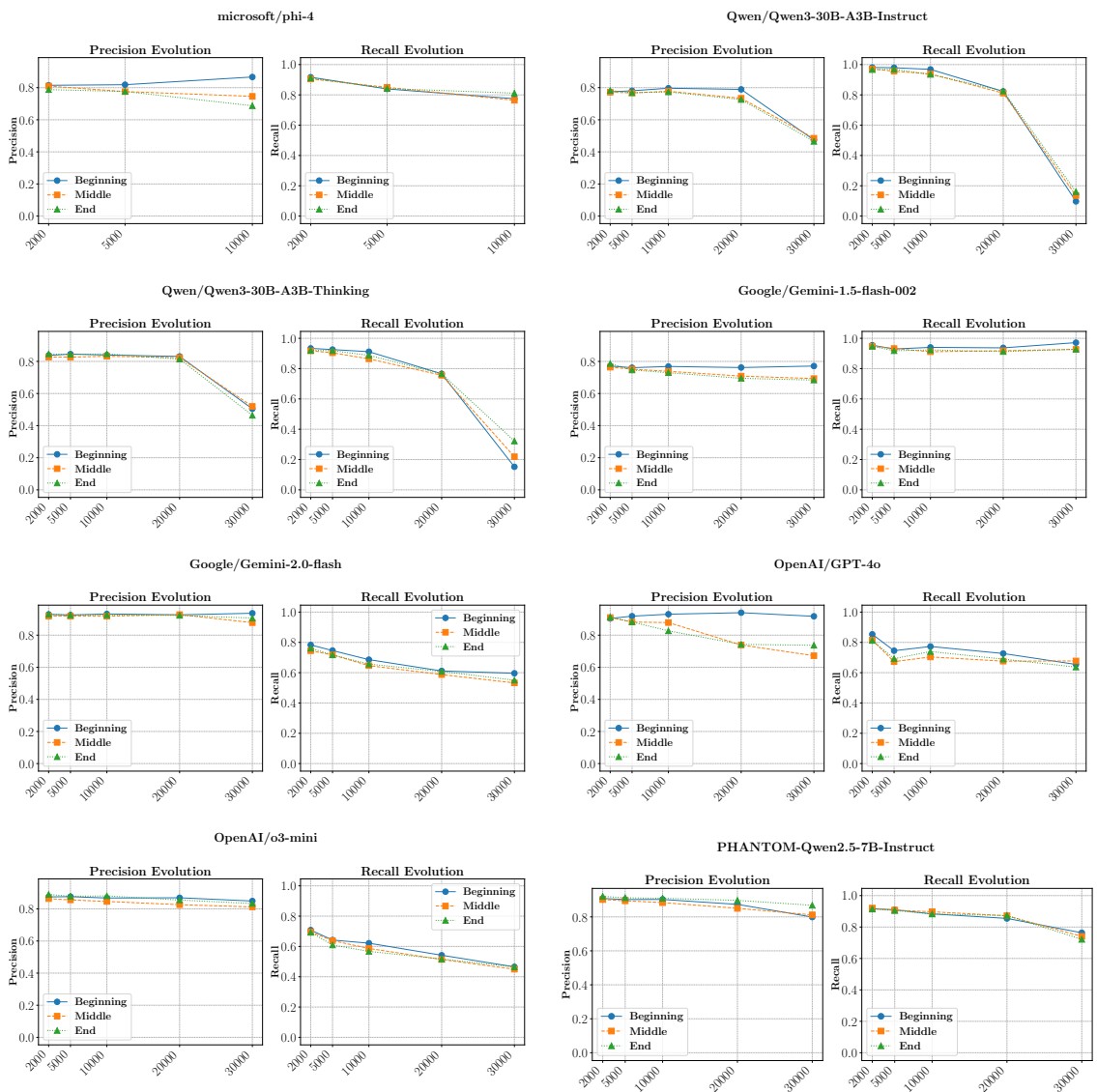

Figure 5: Precision and recall on the long-context benchmark for various closed-source and open-weight models. The x-axis represents the context length in tokens.

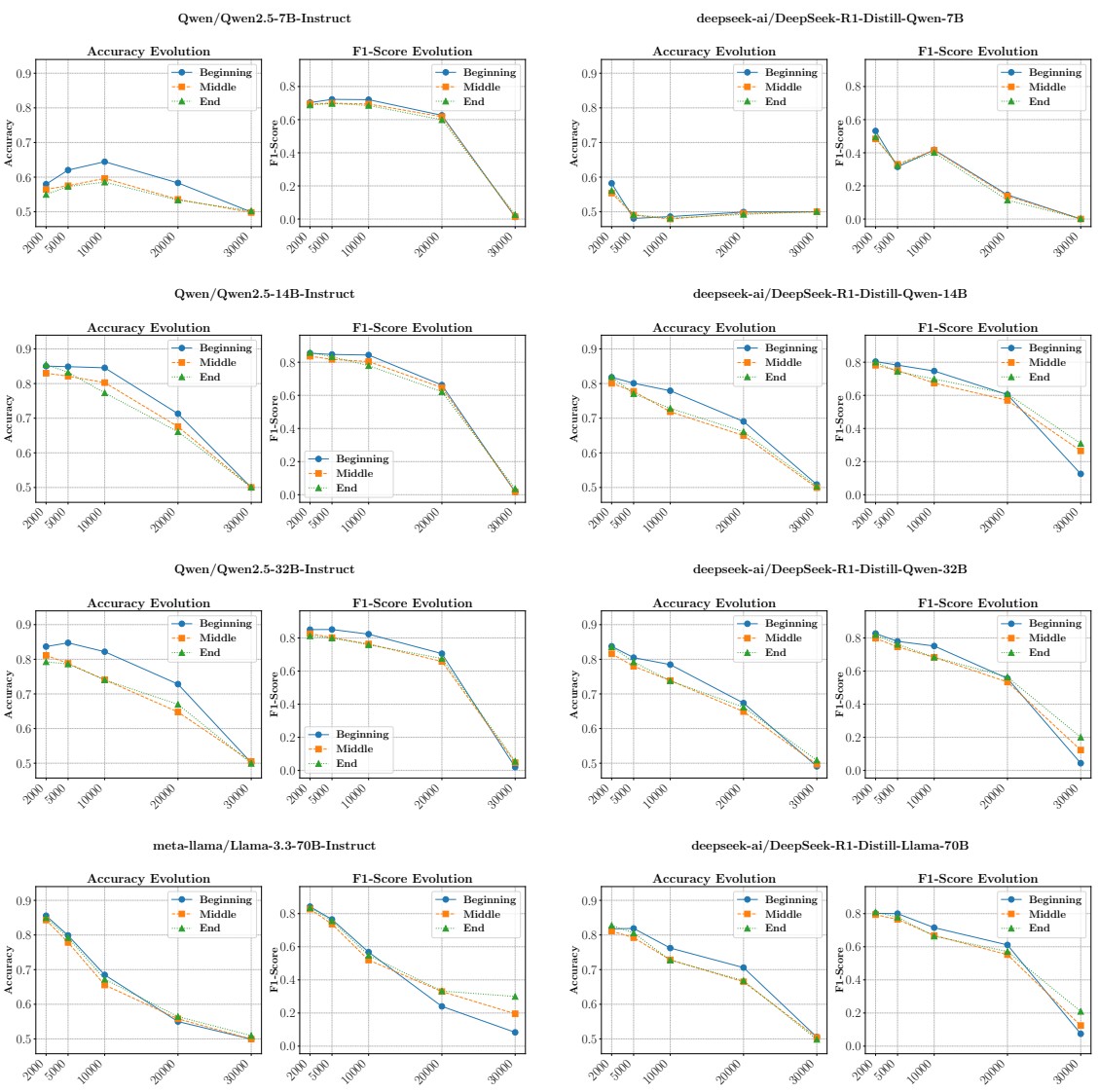

Figure 6: Accuracy and F1 score on the long-context benchmark for base models (left column) and their corresponding distilled versions (right column). The x-axis represents the context length in tokens.

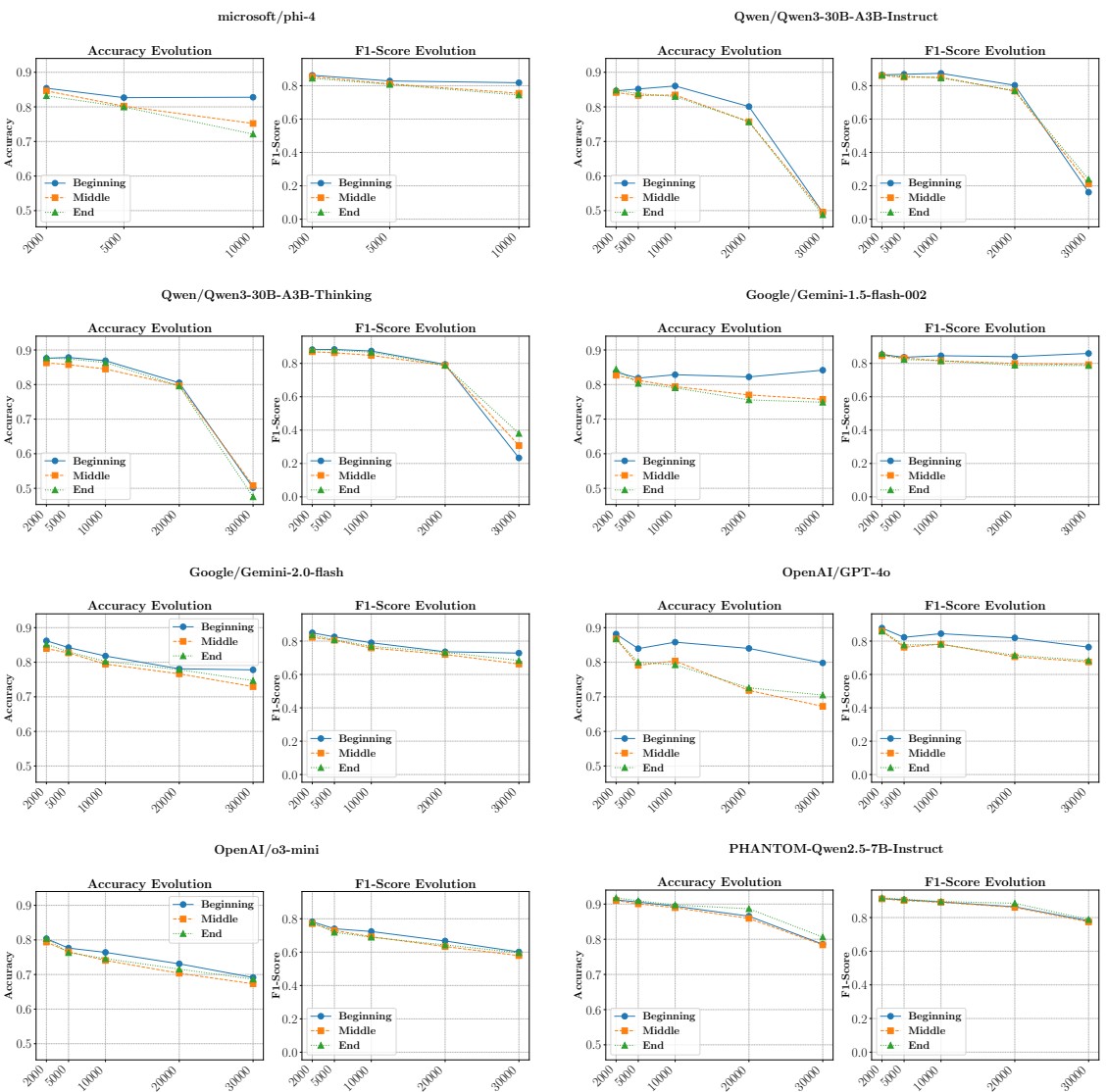

Figure 7: Accuracy and F1 score on the long-context benchmark for various closed-source and open-weight models. The x-axis represents the context length in tokens.

**Query**: What is the total amount of gross proceeds the Company received from issuing and selling Preferred Stock to Dye Cann II?

**Answer**: The Company received aggregate gross proceeds of $21,350,000 from issuing and selling Preferred Stock to Dye Cann II, with the majority of the shares being issued in multiple tranches between December 2020 and March 2021, and the proceeds were primarily used to drive business growth and expansion, including potential investments in emerging markets.

**Ground truth Label**: hallucination

---

**Qwen2.5-7B-Instruct Label**:
not hallucination

**Reasoning Trace**:
REASONING:
- The CONTEXT states that "The Company issued and sold to Dye Cann II 7,700 shares of Preferred Stock on December 16, 2020, 1,450 shares of Preferred Stock on December 18, 2020, 1,300 shares of Series Preferred Stock on December 22, 2020, 3,100 shares of Preferred Stock on February 3, 2021, 1,300 shares of Preferred Stock on February 25, 2021, 2,500 shares of Preferred Stock on March 2, 2021 and 4,000 shares of Preferred Stock on March 30, 2021. As a result, the Company issued and sold an aggregate of 21,350 shares of Preferred Stock to Dye Cann II for aggregate gross proceeds of $21,350,000.

- The answer correctly identifies and summarizes the total gross proceeds from the sale of Preferred Stock to Dye Cann II, which matches the CONTEXT exactly.

- There are no discrepancies or unsupported claims in the ANSWER compared to the CONTEXT.

- The ANSWER is faithful to the CONTEXT and does not introduce any new information or make unsupported claims.

SCORE: PASS

**PHANTOM Qwen2.5-7B-Instruct Label**:
hallucination

**Reasoning Trace**:
{
"REASONING": [
"The ANSWER states that the Company received aggregate gross proceeds of $21,350,000 from issuing and selling Preferred Stock to Dye Cann II. This information is supported by the CONTEXT, which mentions that the Company issued and sold 7,700 shares of Preferred Stock to Dye Cann II for aggregate gross proceeds of $21,350,000.",

"The ANSWER mentions that the majority of the shares were issued in multiple tranches between December 2020 and March 2021. The CONTEXT supports this statement, detailing the specific dates and amounts of shares issued to Dye Cann II.",

"The ANSWER suggests that the proceeds were primarily used to drive business growth and expansion, including potential investments in emerging markets. **The CONTEXT does not explicitly state how the proceeds from the sale of Preferred Stock to Dye Cann II were used**. Therefore, this part of the ANSWER introduces information not present in the CONTEXT, constituting a hallucination."],

"SCORE": "FAIL"
}

Figure 8: Comparsion of Reasoning trace for Qwen2.5-7B-Instruct and PHANTOM-Qwen2.5-7B-Instruct model for 30k context size

