# OpenReview forum: "PHANTOM: A Benchmark for Hallucination Detection in Financial Long-Context QA"
_NeurIPS.cc/2025/Datasets_and_Benchmarks_Track — NeurIPS 2025 Datasets and Benchmarks Track poster_

### Official Review · Reviewer_P3xa · 2025-06-30

**Rating:** 5
**Confidence:** 4

**Summary:**

This paper aims to identify hallucination of LLM in long-context documents in the financial area, which has a high requirement for risk management. And a novel benchmark dataset called PHANTOM for evaluating hallucination detection in long-context financial QA has been proposed and introduced. With detailed experimental testing and evaluation, the proposed benchmark gives insights into how general LLMs’ ability to detect hallucination.

**Additional Feedback:**

1.	As shown in Figure 1(a), the dataset has class imbalance. Does this issue have an impact on learning efficiency? If yes, I think the author should explain the reason.

**Dataset Code Accessibility:**

Yes

**Dataset Code Comments:**

1.	The dataset is accessible in the offered Hugging Face account with a detailed introduction about basic information and a clear guidance for usage.
2.	The code for the data generation and testing is also given, and the experiment design and testing are runnable and explainable.

**Ethical Considerations:**

No, there are no or only very minor ethics concerns

**Final Justification:**

I have read the author's response, and the author carefully explained my question and addressed my concern. I decide to raise the score if the author can highlight these explanations in the paper.

**Limitations Weaknesses:**

1.	In abstract, the author mentions that existing hallucination benchmarks fail to capture the complexities of financial benchmarks, what exactly the “complexities” is? I only see that there are three classes in this dataset, contradiction, misrepresentation, and unverifiable information. I think the author should explain which part can reflect the “complexities” for better understanding.
2.	Why did you choose the context in the area of finance? Since there is also high-risk information in areas like news, online reviews, and so on. The author should also clarify what the difference is between hallucination in financial data with other areas’ data? Thus, the novelty of this work can be strengthened.
3.	In Figure 1(b), the distribution of financial themes in datasets is different; does this variance in distribution affect the learning accuracy?
4.  In addition, the dataset is generated via language model, although human annotators have validated it and subsequently expanded it to capture various context lengths and information placements, it would be better to collect the data from real-world financial-related text.

**Strengths Contributions:**

1.	The proposed PHANTOM dataset targets the hallucination detection in the financial domain, and offers strong support for the LLM-based hallucination detection methods model for learning and testing.
2.	The proposed benchmark dataset focuses on the long context QA, which is challenging in handling hallucination. This work also provides an available resource for addressing long context QA.

---

> ### Author Rebuttal · Authors · 2025-07-31
>
> We thank the reviewer for their thorough evaluation and constructive feedback. We address each concern systematically below.
>
> ## Regarding explanation of “complexities” in financial benchmarks:
> An important detail of our work is that the three hallucination classes were not pre-defined categories that we targeted during generation. Rather, they are the result of a post-hoc analysis of the dataset, as described in Section 4.1. Our generation process was designed to create hallucinations that are ‘minor’, ‘subtle’ and ‘hard to catch’ to mimic the most challenging real-world failure modes. The resulting distribution – where misrepresentation (class 2) constitutes the vast majority – is a finding of our study. We believe this directly reflects the nature of the financial domain.
> We believe that the following non-exhaustive list outlines complexities that arise when dealing with financial benchmarks (there might be overlap with non-financial benchmarks):
>
> - High numerical precision requirements: Financial data demands exact figures where small errors have significant consequences (e.g., $21.35M vs. $21.53M).
>
> - Domain-specific terminology: Terms like "preferred stock" and "aggregate gross proceeds" require specialized understanding.
>
> - Long-context challenges: SEC filings average 30,000+ tokens, creating "lost in the middle" problems not addressed by existing short-context benchmarks.
>
> While designing our prompts for data generation using LLMs, we ensured that these complexities are introduced as part of the hallucinations.
>
> ## Regarding why the authors chose finance as the area of focus of this work:
> We believe that several factors make finance uniquely critical and challenging domain for hallucination research:
>
> - High-stakes consequences: Regulatory compliance with legal implications; direct impact on investment decisions.
>
> - Public data availability: SEC filings provide standardized, authoritative data sources.
>
> - Research opportunity: No prior work addresses long-context financial hallucination detection with systematic methodology.
>
> - Research community contribution: Few datasets are publicly available for financial research. To the best of our knowledge, none is available for hallucination detection for long-context scenarios.
>
> Unique hallucination patterns in finance: Financial hallucinations involve quantitative misrepresentations, domain-specific terminology and extensive document lengths, among others, which makes it distinct from news/online review domains.
>
> ## Regarding whether variance in distribution of financial themes (Figure 1b) or hallucination classes (Figure 1a) affects learning accuracy:
> As we note in Section 4.1, the thematic distribution is a direct result of our data sourcing from different types of SEC filings. By preserving this natural distribution, PHANTOM provides a more realistic and ecologically valid benchmark, testing LLMs abilities to handle varied and imbalance topical landscape they would encounter in practical applications.
>
> To address the reviewer’s concern about learning accuracy, our experimental results show that model performance remains robust despite thematic variance. For instance, our strongest baseline model, Llama-3.3-70B-Instruct achieves consistently high F1-scores across varied datasets (Table 7). Furthermore, our finetuned model (last row in Table 7) performs about equally well on 10-K and DEF 14A datasets, thus providing more evidence that thematic differences in the training have minimal impact on the learning accuracy.
>
> ## Regarding synthetic dataset generation and whether it would be better to collect data from real-world financial-related text:
> Our dataset is constructed from real-world financial documents, specifically public SEC filings. While question-answer pairs and hallucinations are generated via an LLM, they are directly based on content from these authentic documents. This semi-synthetic approach grounds our benchmark in the complex, domain-specific language of finance, ensuring practical relevance. Furthermore, this methodology is consistent with other established benchmarks in NLP that use synthetic generation to evaluate complex reasoning and faithfulness tasks where labeled data is scares, such as HaluEval (Li et al. (2023)) and ARES (Saad-Falcon et al. (2024)). Further, we wish to elaborate on the methodological choices we made to use LLM generated answers and hallucinated answers:
>
> - Addressing a critical data bottleneck in financial AI: we acknowledge in our limitations section (lines 360-366) that a synthetically generated dataset may not capture every nuance of real-world hallucinations. However, our work directly addresses a major bottleneck in the field: the extreme scarcity of large-scale, labeled datasets for financial hallucination detection. PHANTOM’s contribution is therefore not just the dataset itself, but a replicable and scalable methodology for creating high-quality, class-balanced long context dataset in a domain where it is otherwise infeasible to obtain.
>
> - Grounded, not purely synthetic generation: we ensure realism by grounding our data generation in real documents and examples. Each data point is constructed on a contiguous text chunk from an SEC filing (lines 146-151, 168-169). Hallucination generation was guided by instructions and real-world examples of subtle failures observed in production AI systems (lines 181-184), ensuring they mimic realistic, nuanced errors in real data.
>
> - Enabling controlled experimental design: a primary motivation for our synthetic approach is to enable experiments that are impossible with naturally occurring data. As highlighted in lines 72-75, PHANTOM’s design allows the exact same query and answer to be tested across various context lengths and positions of information. This enables a controlled experiment setting that cannot be achieved with disconnected, real-world data points.
>
> Li et al. (2023). “HaluEval: A Large-Scale Hallucination Evaluation Benchmark for Large Language Models”. EMLP.
>
> Saad-Falcon et al. (2024). “ARES: An Automated Evaluation Framework for Retrieval-Augmented Generation Systems”. NAACL.

---

### Official Review · Reviewer_qLBJ · 2025-07-01

**Rating:** 5
**Confidence:** 4

**Summary:**

The paper sets out to create a benchmark for hallucination specifically in Finance domain.
It starts with a "seed dataset" with query-answer-chunk triplets and then uses LLM to generate hallucinations (verified by humans on a subset) - and then the data context is augmented by going back to the original source document, and putting the golden chunk in different positions -- but maintaining the original contextual flow and order.

Different models are then evaluated in the the newly created Bencmark (PHANTOM) on different financial document types. Effect on increasing context length is also investigated.

**Additional Feedback:**

> We evaluate the finetuned model, which we call PHANTOM-Qwen2.5-7B-Instruct, on 10-K and DEF
339 14A datasets (we omit 8-K and 497K because these datasets were involved in the training process).

* Are there dedicated train/dev/test splits per document type? If so, I think the evalaution on 8-K and 497K 339 can still be included - if there is no leakage to some test set associated with them.

**Dataset Code Accessibility:**

Yes

**Dataset Code Comments:**

There is a huggingface link which can be signed into.

**Ethical Considerations:**

No, there are no or only very minor ethics concerns

**Final Justification:**

I maintain my score as nothing fundamental have changed to change my original decision either way.
One positive addition from the authors is the evaluation on more propriety models - however, that's not groundbreaking enough for 6.

**Limitations Weaknesses:**

I don't have any glaring technical issues with the paper. The authors themselves have an exhaustive limitation section as well.

Few minor points:

1. Some benchmarking of popular closed-source models (gpt4o, gemini etc.) could have been good but not critical for acceptance.
1. One analyses I would have been interested in, is how much of a different the model used to generate hallucination can make. I would be curious if there's some confounding factor that makes LLAMA 3.3 to be particularly good in detecting the hallucinations because it is the same model that generated them. One investigation can be to test (possibly on a small subset of the seed dataset) using an alternative hallucination generator - like deepseek distill -- after that would deepseek become the best model or llama 3.3 remains the best?

**Strengths Contributions:**

* Overall, a highly readable clean targeted paper.
* Tackles an important tasks (detecting hallucination) for a relevant domain (Finance).
* The dataset can be very useful for future research.
* Shows the benefit of fine-tuning for not just improved performance in standard settings but also better robustness to increased length.
* Good limitation section.

---

> ### Author Rebuttal · Authors · 2025-07-31
>
> Thank you for your thorough and constructive review of our paper "PHANTOM: A Benchmark for Hallucination Detection in Financial Long-Context QA." We are pleased that you found our contribution valuable and appreciate your positive assessment. We address your specific concerns and suggestions below.
>
> ## Regarding benchmarking closed-source models:
> While the focus of the paper is on generating synthetic datasets that can be used for benchmarking at scale, we do wish to emphasize that the PHANTOM datasets can be used to benchmark any commercial models. The results are not published here due to confidentiality obligations (we will add a disclaimer in the final version of the paper if accepted). We hope that releasing this dataset publicly will enable other researchers to benchmark closed-source models in the future.
>
> ## Regarding using a different model than Llama 3.3 to generate hallucinations:
> Our selection of Llama-3.3-70B-Instruct was based on its standing as one of the state-of-the-art open-source models at the time of our dataset’s creation.
>
> We agree that using different models for generation is an interesting direction for future work to enhance dataset diversity. For this study, however, we prioritized diversity in the source material as mentioned above, which we consider more fundamental for a robust financial benchmark.
>
> ## Regarding train/dev/test splits:
> The PHANTOM dataset is designed to facilitate robust evaluation by separating training and testing data to ensure out-of-sample testing. As detailed in Section 5.1 of the document, the fine-tuning of the PHANTOM-Qwen2.5-7B-Instruct model was performed using training data generated from the Form 8-K and Form 497K seed datasets. To ensure a true out-of-sample test and avoid data leakage, the evaluation of this fine-tuned model was explicitly conducted on the 10-K and DEF 14A datasets, while the 8-K and 497K datasets were omitted from this particular evaluation, as stated in Section 5.2. This strategy allows for a clear assessment of the model's generalization capabilities on unseen financial document types. Appendix E (Table 7) further illustrates the evaluation results, showing 'N/A' for 8-K and 497-K filings for the PHANTOM-Qwen2.5-7B-Instruct model, consistent with their use in the training phase rather than the evaluation phase for this specific fine-tuned model.

---

> > ### Comment · Reviewer_qLBJ · 2025-08-02
> >
> > > The PHANTOM dataset is designed to facilitate robust evaluation by separating training and testing data to ensure out-of-sample testing. As detailed in Section 5.1 of the document, the fine-tuning of the PHANTOM-Qwen2.5-7B-Instruct model was performed using training data generated from the Form 8-K and Form 497K seed datasets. To ensure a true out-of-sample test and avoid data leakage, the evaluation of this fine-tuned model was explicitly conducted on the 10-K and DEF 14A datasets, while the 8-K and 497K datasets were omitted from this particular evaluation, as stated in Section 5.2. This strategy allows for a clear assessment of the model's generalization capabilities on unseen financial document types. Appendix E (Table 7) further illustrates the evaluation results, showing 'N/A' for 8-K and 497-K filings for the PHANTOM-Qwen2.5-7B-Instruct model, consistent with their use in the training phase rather than the evaluation phase for this specific fine-tuned model.
> >
> > On this point, I meant that the OOD performance can be evaluation without necessarily sacrificing IID performance evaluation, if the test sets are available for all the relevant stratifications (both OOD splits, and IID splits).

---

> > > ### Author Response · Authors · 2025-08-06
> > >
> > > Thank you for the clarification and for your continued insightful feedback. We greatly appreciate your engagement during the discussion period.
> > >
> > > Your point is very well-taken. We agree completely that evaluating both IID and OOD performance would provide a more comprehensive and valuable assessment of the model. Our initial experimental design focuses on a strict OOD-only evaluation for the fine-tuned model to emphasize its ability to generalize. However, you are absolutely right that by creating held-out test sets for the 8-K and 497K filings, we could have simultaneously evaluated IID performance without compromising the OOD analysis.
> > >
> > > This is an excellent suggestion for improvement. We will update Section 5.2 to include a more complete evaluation scheme, including creating train/test splits for all filing types, allowing us (also future work) to benchmark and provide results on both IID and OOD performance.

---

### Official Review · Reviewer_aMbJ · 2025-07-02

**Rating:** 3
**Confidence:** 4

**Summary:**

This paper introduces PHANTOM, a novel benchmark dataset for evaluating hallucination detection in long-context financial question answering. Built using SEC filings and validated through both LLMs and human annotators, PHANTOM allows controlled evaluations across varying context lengths (up to 30K tokens) and information positions (beginning, middle, end). The authors demonstrate the utility of PHANTOM through benchmarking and fine-tuning experiments. However, this work has some deficiencies in the construction of the dataset and the experimental evaluation.

**Additional Feedback:**

1. Regarding Lines 152–157, please clarify the proportional distribution within the dataset.
2. Given the sensitivity of financial data, please clarify whether the data obtained from the Securities and Exchange Commission (SEC) was used with proper authorization or under an appropriate license.

**I would raise the overall score if the aforementioned concerns and limitations are adequately addressed.**

**Dataset Code Accessibility:**

Yes

**Dataset Code Comments:**

The authors have provided private GitHub and Hugging Face account credentials in the supplementary materials, which can be used to access the code and datasets upon logging in.

**Ethical Comments:**

No ethical concerns.

**Ethical Considerations:**

No, there are no or only very minor ethics concerns

**Final Justification:**

While the authors' response addresses most of my concerns, I maintain reservations regarding the acceptance of this paper. This is primarily due to the substantial volume of additional experimentation required (nearly doubling the original work), which has resulted in an insufficient evaluation within the originally submitted manuscript. Furthermore, potential ethical concerns necessitate examination by an ethics reviewer. Nevertheless, as a response to the author's comprehensive rebuttal, I have upgraded my score.

**Limitations Weaknesses:**

> Limitations of Seed Dataset Generation.

1. Critical questions arise regarding how to ensure the diversity and randomness of the data, guarantee the reliability of LLM-based faithful answer generation, and substantiate the rationale and diversity of hallucinations in LLM-based hallucinated answer generation.
2. Furthermore, the justification for employing the Llama-3-70B-Instruct model requires clarification, including whether comparative analyses were conducted with other LLMs or advanced reasoning LLMs regarding their performance.
3. Relying solely on manual verification of 10% of the generated data is insufficient to address these fundamental concerns.

> Types of Hallucinations.

1. "Class 1 - The answer to the question contradicts the provided context. Class 2- The answer to the question misrepresents the provided context." What is the distinction between these two types of hallucinations? Could illustrative examples be provided?
2. What is the correspondence between the three hallucination types defined in this paper and the commonly recognized categories of factuality hallucination and faithfulness hallucination?
3. More critically, Lines 117-119 mention that "the specific linguistic and logical complexities inherent in financial documents influence the manifestation of hallucinations." However, how are the three defined hallucination types specifically related to financial data or the challenges of long financial contexts? It appears these three types do not distinctly address hallucination issues unique to financial data. Instead, they seem predominantly defined based on the QA task setup.

> Quality of Benchmark.

As a hallucination benchmark, the hallucinations were induced solely by a single LLM, and the hallucination types employed are oversimplified. This benchmark fails to adequately highlight the unique challenges inherent in financial data and long-context financial scenarios. Consequently, existing models did not exhibit particularly poor performance on this benchmark. For instance, even the most basic model evaluated in the experiments, Qwen2.5-7B-Instruct, achieved an accuracy close to 60%. This high baseline performance makes it difficult for the benchmark to fully assess the performance gap between more advanced baseline models.

> Lack of comprehensive experimental evaluation.

1. The evaluation is limited to only three series of baseline models (albeit including variants of different sizes or types), which is far from sufficient for a benchmark work. While the authors pose three questions in Line 287, each is validated using only a single series of models, lacking representativeness. For instance, why were open-source reasoning models like the Qwen3 series (released before the submission deadline) not evaluated?
2. During the LLM-based data construction phase, Llama-3-70B-Instruct was employed.  Directly using this model to evaluate hallucination detection performance is methodologically unsound, as it tends to perform well on the data it generates itself. Relying solely on this single "SOTA model with large parameter size" (RQ1 in Line 289) renders the evaluation severely insufficient.

**Strengths Contributions:**

1. This paper targets hallucination in financial long-context QA—a high-stakes yet under-explored domain.
2. The seed-dataset expansion strategy (Section 3.2) ensures ecological validity and efficiency, avoiding pitfalls of synthetic document concatenation. The benchmark generation methodology maintains contextual realism via contiguous SEC filing extensions.
3. Fine-tuning on PHANTOM boosts small models, enabling cost-efficient deployment.

---

> ### Author Rebuttal · Authors · 2025-07-31
>
> We thank the reviewer for their detailed feedback and constructive criticism.
> # Addressing Limitations of Seed Dataset Generation
> 1. Regarding "data diversity and randomness”:
> - Document structure and purpose: to capture a variety of financial reporting styles, terminologies, diverse companies and industries, and structures, we deliberately sourced data from four of the most common but functionally and structurally distinct types of SEC filings, detailed in lines 146-157.
> - Random chunk extraction: to ensure randomness within diverse sources, the initial “seed chunk” was extracted randomly from full filing documents (lines 168-169). This prevents systematic bias in content selection.
>
> Regarding “reliability of LLM-based answer generation”:
>
> We agree that ensuring reliability of a benchmark dataset with synthetically generated elements is paramount. We have expanded our manual validation to 38.4% subset of the seed dataset. Further, we made the choice to use LLM generated answers and hallucinated answers for the reasons of:
> - Addressing a critical data bottleneck in finance: Our work directly addresses a major bottleneck in the field: the extreme scarcity of large-scale, labeled datasets for financial hallucination detection. PHANTOM’s contribution is a replicable and scalable methodology for creating high-quality, class-balanced long context dataset in a domain where it is otherwise infeasible to obtain.
> - Grounded, not purely synthetic generation: we ground our data generation in real documents and examples. Each data point is constructed on a real text chunk from an SEC filing (lines 146-151, 168-169). Hallucination generation was guided by instructions and real-world examples of subtle failures observed in production AI systems (lines 181-184).
> - Enabling controlled experimental design: a primary motivation of our synthetic approach is to enable experiments that are impossible with naturally occurring data. As highlighted in lines 72-75, PHANTOM’s design allows the exact same query and answer to be tested across various context lengths and positions of information.
>
> 2. Regarding  “justification for employing Llama-3 model”, our selection of the model was based on its standing as one of the SOTA open-source models at the time of our dataset’s creation. We agree that using different models for generation is an interesting direction to potentially enhance dataset diversity. Our goal was to create a high-quality dataset (substantiated through manual validation, evaluation and fine-tuning results). An elaborate study on data generation capabilities of various LLMs can definitely be an important future work.
>
> 3. Regarding  “10% manual verification”, our approach of combining synthetic data generation with a smaller human-annotated set is a common, accepted methodology in the field, used by benchmarks like HaluEval (Li et al. (2023)) (14.2% validation) and RAGEval (Zhu et al. (2024)) (9.3% validation).
> Also, we have expanded human validation to 38.4% of the seed data. In total we reviewed 1,543 out of 3,962 samples, with 21 being labeled as incorrect and 98.6% of samples confirmed to be correct, which is comparable to numbers reported in the paper. We plan to accordingly update the final version of the paper (if accepted).
>
> # Addressing Concerns on Types of Hallucinations:
> 1. Regarding your question “the distinction between the two types of hallucinations (Class 1 and class 2)”:
> - Class 1 Contradiction: The answer directly conflicts with information explicitly stated in the provided context.
> - Class 2 Misrepresentation: The answer is partially based on the context but adds significant, unsupported details or assertions, resulting in distortion or embellishment of facts present in the text.
>
> ## Example - Class 1:
> Query: What is the retirement policy for Independent Trustees in the Eaton Vance fund complex?
> Relevant snippet from Context containing the answer: ... Under the terms of each Fund’s current Trustee retirement policy, an Independent Trustee must retire and resign as a Trustee on the earlier of: (i) the first day of July following his or her 74th birthday;
>
> Hallucinated Answer: Under the terms of each Fund’s current Trustee retirement policy, an Independent Trustee must retire and resign as a Trustee on the earlier of: (i) the first day of July following his or her 75th birthday;
> Reasoning for classification: This is a Class 1 Contradiction because the answer contains a factual error. It incorrectly states the retirement age for Independent Trustees is 75 (instead of 74), directly contradicting the context.
>
> ## Example - Class 2:
> Query: What percentage of the Company's total purchases were accounted for by its two largest vendors during the years ended December 31, 2023 and 2022?
> Relevant snippet from Context containing the answer: ...At December 31, 2023 and 2022, all of the Company's cash and cash equivalents were held at one accredited financial institution. The Company is dependent on third-party vendors for its inventory purchases. During the years ended December 31, 2023 and 2022, two vendors accounted for 35% and 59% of total purchases, respectively...
>
> Hallucinated Answer: During the years ended December 31, 2023 and 2022, the Company's two largest vendors accounted for approximately 35% and 59% of total purchases, respectively, with these vendors primarily supplying inventory for the Company's core operations, which has been a consistent trend over the past few years.
> Reasoning for classification: The answer is "Class 2 Misrepresentation" because it is partially based on the context, but it made an assertion that was not supported by the provided context. While it correctly reported vendor purchase percentages for 2022 and 2023, it misrepresented this as a "consistent trend."
>
> 2. Regarding your question about the correspondence between the three hallucination classes and commonly recognized categories: Our classes align with "source faithfulness" or "intrinsic hallucination," focusing exclusively on detecting hallucinations relative to the provided context, similar to prior work (e.g., Li, et al. (2024)). We contend that our system offers a more granular categorization within intrinsic hallucination, which is crucial for financial applications where even subtle misrepresentations (Class 2) can be as problematic as direct contradictions (Class 1).
>
> 3.The three identified hallucination classes were derived from a post-hoc analysis of generated data, rather than being predefined categories. The study aimed to create minor, subtle, and hard-to-catch hallucinations to emulate challenging real-world failure modes, particularly in the financial domain, where misrepresentation (class 2) was found to be the predominant type of hallucination.
>
> # Addressing Concerns on Quality of Benchmark:
> Regarding ‘...the most basic model Qwen2.5-7B-Instruct, achieved an accuracy close to 60%. This high baseline performance ... ‘, we argue that the ~60% accuracy on Qwen2.5-7B-Instruct demonstrates appropriate benchmark difficulty. Recent benchmark analysis shows that effective benchmarks should allow smaller models to perform above random (50% for binary classification, which is the case for our benchmark) while still challenging larger models (Li, et al. (2025)). Our results show clear performance differentiation across model sizes, with substantial degradation in long-context scenarios - exactly the phenomenon we aimed to capture.
>
> # Addressing Lack of comprehensive experimental evaluation
> | Model | Accuracy | Precision | Recall | F1 Score |
> |---|---|---|---|---|
> Phi4 | 0.874 | 0.816| 0.967 | 0.885 |
> | Deepseek R1 | 0.907 | 0.869 | 0.960 | 0.912 |
> | Vectara HHEM-2.1-Open | 0.639 | 0.594 | 0.884 | 0.712 |
>
> 1. On the scope of the experimental evaluation
> We agree that a broad evaluation is important for a benchmark paper. While the focus of this paper is on generating synthetic datasets that can be used for benchmarking at scale, we wish to emphasize that the PHANTOM datasets can be used to benchmark any commercial models. The results on commercial models are not published due to
> confidentiality obligations.
> Regarding newer models like Qwen 3, it was released on 04/28/2025. Our experimental phase was conducted prior to this release, which is why Qwen 2.5 was used. We are also expanding our evaluation and providing results on the seed dataset for Phi-4, DeepSeek R1 and Vectara (Bao et al. (2024)). We also commit to including long-context results for these models and Qwen3 in the camera-ready version.
>
> 2. On the methodological soundness of evaluating the data-generating model
> We deliberately evaluated Llama-3.3 to investigate if it held an unfair advantage. Our results showed that there isn’t a clear sign of that. Llama-3.3 did not achieve perfect scores, and its performance degrades significantly on long-context tasks (Figure 2), falling into a similar range as other models. We believe this result shows that PHANTOM (long-context) dataset remains a challenging benchmark even for the data-generating model. Further, this practice is standard in benchmark creation, provided two conditions are met: 1) there is a reliable human validation process and 2) a diverse set of other models are evaluated for comparison (Li et al. (2023)). Our work satisfies both.
>
> Additional Feedback: 1.See Appendix B Dataset Statistics.2.See response to reviewer Bnf2.
> # References:
> Bao et al. (2024). “HHEM-2.1-Open”. Hugging Face Library: https://huggingface.co/vectara/hallucination_evaluation_model
>
> Li, et al. (2025). "VL-RewardBench: A Challenging Benchmark for Vision-Language Generative Reward Models." CVPR.
>
> Li et al. (2023). “HaluEval: A Large-Scale Hallucination Evaluation Benchmark for Large Language Models”. EMLP.
>
> Zhu, K et al. (2024). “RAGEval: Scenario Specific RAG Evaluation Dataset Generation Framework”. arXiv.
>
> Li, et al. (2024) "The Dawn After the Dark: An Empirical Study on Factuality Hallucination in Large Language Models." ACL.

---

> > ### Comment · Reviewer_aMbJ · 2025-08-04
> >
> > Thank you for your response, which has partially addressed my concerns. I will consider raising my score. However, I maintain my decision leaning toward rejection due to the lack of comprehensive experimental evaluation. As a benchmark, the existing evaluation is far from sufficient, and critically, it does not evaluate against any powerful closed-source models.

---

> > > ### Author Response · Authors · 2025-08-06
> > >
> > > Thank you for your thoughtful feedback and your willingness to continue the discussion and reconsider your score. We sincerely appreciate your perspective on the importance of a comprehensive experimental evaluation for a benchmark paper.
> > >
> > > We want to respectfully acknowledge the validity of your point; in an ideal scenario, a benchmark evaluation would indeed include results from powerful closed-source models to provide the broadest possible spectrum of performance evaluation.
> > >
> > > ### We have tested closed-source models on our benchmark. And as stated in our rebuttal, we do wish to emphasize that PHANTOM can be used to benchmark any commercial models. However, the exclusion of these models from our paper is not a matter of choice, but a legal constraint imposed by the confidentiality obligations associated with using these commercial products. This is a common and significant challenge for industry researchers wishing to contribute to public benchmarks. Publishing such results would unfortunately place us in violation of these terms.
> > >
> > >
> > > Given this constraint, we wish to re-emphasize the core contributions of our paper:
> > >
> > > 1. **The Primary Contribution is the Publicly Available Benchmark**: Our main goal is to provide the research community with PHANTOM itself – a novel, carefully constructed, and high-quality dataset for a critical and under-explored area. Our work addresses a critical bottleneck in the field: the extreme scarcity of large-scale, labeled datasets for financial hallucination detection.
> > >
> > > 2. **Establishing a Reproducible baseline**: The experiments included in our paper serve to establish a robust and, crucially, fully reproducible baseline using state-of-the-art open-source models. By focusing on open-source models, for both generation and evaluation, we ensure that any researcher can verify our findings and use our results as a reliable starting point of their own work. This reproducibility is a key strength that is not possible with closed-source models, whose performance can vary over time.
> > >
> > > 3. **Demonstrating the Benchmark’s Utility**: Our evaluation, while focused on open source models, effectively demonstrates PHANTOM’s utility. It successfully highlights the severe challenges current LLMs face with long-context hallucination detection, quantifies the “lost in the middle” problem, and shows how our dataset can be used to drive improvements, as seen in our fine-tuning experiments.
> > >
> > >
> > > Ultimately, our work addresses a critical and demonstrable void in the current landscape. We introduced PHANTOM because there were no existing benchmarks specifically designed to tackle the nuanced challenge of hallucination within long-context financial documents. This is not just an incremental addition, but a foundational resource for a high-stakes domain where errors can have significant consequences. By providing PHANTOM, we open new avenues for research, empowering the community – from academic to industry labs – to rigorously benchmark mitigation strategies, develop new detection models, and evaluate any system, whether open-source or proprietary (for those not bound by the same confidentiality constraints.)
> > >
> > > We are fully committed to expanding the open-source evaluation as promised in our rebuttal (including Qwen3, Phi-4, DeepSeek R1, HHEM-2.1-Open across all long-context settings). And hope you will agree that providing the community with this challenging and reproducible benchmark is a valuable contribution that will encourage essential future work.
> > >
> > > Thank you again for your constructive engagement.

---

> > > > ### Comment · Reviewer_aMbJ · 2025-08-06
> > > >
> > > > Thanks for the authors' response. I wish to clarify that my assessment of this work carries no personal bias or prejudice. My inquiry stems solely from a desire to understand:
> > > >
> > > > Why are the evaluation results of proprietary models not disclosed? What specific confidentiality obligations or legal constraints prevent the authors from providing these results?
> > > >
> > > > In my view, since the authors propose an open-source financial hallucination dataset, they bear an obligation to enable evaluations using various popular state-of-the-art (SOTA) models. This is fundamental to any benchmarking study and aligns with standard practices for newly proposed benchmarks.
> > > >
> > > > Unless the authors explicitly demonstrate that disclosing results from closed-source models violates specific confidentiality agreements or statutory regulations, this omission remains problematic. Should such legal barriers exist, however, it would raise potential ethical concerns regarding the dataset itself.
> > > >
> > > > I regret that this raises further concerns, which may disappoint the authors, but I trust they will recognize this as genuine scholarly interest in the work’s integrity—not personal criticism.
> > > >
> > > > Best regards.

---

> > ### Comment · Reviewer_aMbJ · 2025-08-07
> >
> > I look forward to your further response and hope to maintain ongoing communication with you.

---

> > > ### Author Response · Authors · 2025-08-07
> > >
> > > Thanks for helping us to raise the bar for the paper. We understand your concern about the validity of the results on a broader set of models. We met with our legal team and explained the situation further. They have kindly approved our request to release closed-source LLM results in our paper, under the condition that the LLM provider will be able to replicate and validate the results, following our experimental setup, if they need to.
> > >
> > > We are sharing partial results on four closed-sourced models (specifically chosen from the Google and OpenAI families), and we are actively working to complete the evaluation. The following table shows the evaluation results on datasets generated from 10K filing documents. We are committed to completing and sharing this table by the end of the discussion period tomorrow. We will, of course, include the full set of results for both the seed and long-context datasets (including 10k, 20k, 30k token datasets) - across multiple filing types, in the camera-ready version.
> > >
> > >
> > > | Model |   500 tokens seed |   2000 tokens beginning |   2000 tokens middle |   2000 tokens end |   5000 tokens beginning |   5000 tokens middle |   5000 tokens end |
> > > |-------------|-------------------------------------------|-------------------------------------------|---------------------------------------|-------------------------------------|-------------------------------------------|---------------------------------------|-------------------------------------|
> > > | Gemini 1.5 flash 002 | Accuracy: 0.880, Precision: 0.823, Recall: 0.970, F1 score: 0.890 | Accuracy: 0.848, Precision: 0.777, Recall: 0.978, F1 score: 0.866 | Accuracy: 0.849, Precision: 0.784, Recall: 0.966, F1 score: 0.865 | Accuracy: 0.874, Precision: 0.812, Recall: 0.972, F1 score: 0.885 | Accuracy: 0.845, Precision: 0.770, Recall: 0.985, F1 score: 0.864 | Accuracy: 0.823, Precision: 0.752, Recall: 0.964, F1 score: 0.845 | Accuracy: 0.843, Precision: 0.772, Recall: 0.974, F1 score: 0.861 |
> > > | Gemini 2.0 flash | Accuracy: 0.897, Precision: 0.946, Recall: 0.843, F1 score: 0.891 | Accuracy: 0.882, Precision: 0.932, Recall: 0.824, F1 score: 0.875 | Accuracy: 0.851, Precision: 0.931, Recall: 0.758, F1 score: 0.835 | Accuracy: 0.865, Precision: 0.929, Recall: 0.790, F1 score: 0.854 | Accuracy: 0.853, Precision: 0.921, Recall: 0.773, F1 score: 0.840 | Accuracy: 0.848, Precision: 0.924, Recall: 0.758, F1 score: 0.833 | Accuracy: 0.849, Precision: 0.924, Recall: 0.760, F1 score: 0.834 |
> > > | GPT 4o | Accuracy: 0.914, Precision: 0.915, Recall: 0.913, F1 score: 0.914 | Accuracy: 0.901, Precision: 0.921, Recall: 0.877, F1 score: 0.899 | Accuracy: 0.897, Precision: 0.938, Recall: 0.851, F1 score: 0.892 | | | | |
> > > | o3 mini | Accuracy: 0.863, Precision: 0.919, Recall: 0.797, F1 score: 0.853 | Accuracy: 0.847, Precision: 0.889, Recall: 0.794, F1 score: 0.839 | Accuracy: 0.840, Precision: 0.880, Recall: 0.788, F1 score: 0.832 | | | | |
> > >
> > > And finally, we wish to reiterate that the PHANTOM dataset itself is approved to be open-sourced (planned to be released under Apache 2.0 license, which allows commercial and non-commercial usage), so that any researcher can use the dataset to evaluate any model they have access to, without restrictions. We hope to come back with a broader set of results before the end of the discussion period.

---

> > > > ### Author Response · Authors · 2025-08-08
> > > >
> > > > Dear reviewer,
> > > > Following up on our previous message, we have completed the evaluation on the following datasets (up to 5000 token length) generated from 10K filing documents, as promised. Please find the full table of results below.
> > > >
> > > > | Model |   500 tokens seed |   2000 tokens beginning |   2000 tokens middle |   2000 tokens end |   5000 tokens beginning |   5000 tokens middle |   5000 tokens end |
> > > > |-------------|-------------------------------------------|-------------------------------------------|---------------------------------------|-------------------------------------|-------------------------------------------|---------------------------------------|-------------------------------------|
> > > > | Gemini 1.5 flash 002 | Accuracy: 0.880, Precision: 0.823, Recall: 0.970, F1 score: 0.890 | Accuracy: 0.848, Precision: 0.777, Recall: 0.978, F1 score: 0.866 | Accuracy: 0.849, Precision: 0.784, Recall: 0.966, F1 score: 0.865 | Accuracy: 0.874, Precision: 0.812, Recall: 0.972, F1 score: 0.885 | Accuracy: 0.845, Precision: 0.770, Recall: 0.985, F1 score: 0.864 | Accuracy: 0.823, Precision: 0.752, Recall: 0.964, F1 score: 0.845 | Accuracy: 0.843, Precision: 0.772, Recall: 0.974, F1 score: 0.861 |
> > > > | Gemini 2.0 flash | Accuracy: 0.897, Precision: 0.946, Recall: 0.843, F1 score: 0.891 | Accuracy: 0.882, Precision: 0.932, Recall: 0.824, F1 score: 0.875 | Accuracy: 0.851, Precision: 0.931, Recall: 0.758, F1 score: 0.835 | Accuracy: 0.865, Precision: 0.929, Recall: 0.790, F1 score: 0.854 | Accuracy: 0.853, Precision: 0.921, Recall: 0.773, F1 score: 0.840 | Accuracy: 0.848, Precision: 0.924, Recall: 0.758, F1 score: 0.833 | Accuracy: 0.849, Precision: 0.924, Recall: 0.760, F1 score: 0.834 |
> > > > | GPT 4o | Accuracy: 0.914, Precision: 0.915, Recall: 0.913, F1 score: 0.914 | Accuracy: 0.901, Precision: 0.921, Recall: 0.877, F1 score: 0.899 | Accuracy: 0.897, Precision: 0.938, Recall: 0.851, F1 score: 0.892 | Accuracy: 0.888, Precision: 0.917, Recall: 0.853, F1 score: 0.884 | Accuracy: 0.878, Precision: 0.940, Recall: 0.808, F1 score: 0.869 | Accuracy: 0.853, Precision: 0.928, Recall: 0.765, F1 score: 0.839 | Accuracy: 0.873, Precision: 0.920, Recall: 0.817, F1 score: 0.865|
> > > > | o3 mini | Accuracy: 0.863, Precision: 0.919, Recall: 0.797, F1 score: 0.853 | Accuracy: 0.847, Precision: 0.889, Recall: 0.794, F1 score: 0.839 | Accuracy: 0.840, Precision: 0.880, Recall: 0.788, F1 score: 0.832 | Accuracy: 0.838, Precision: 0.905, Recall: 0.756, F1 score: 0.824 | Accuracy: 0.815, Precision: 0.883, Recall: 0.725, F1 score: 0.796| Accuracy: 0.806, Precision: 0.863, Recall: 0.727, F1 score: 0.789| Accuracy: 0.816, Precision: 0.894, Recall: 0.716, F1 score: 0.795|
> > > >
> > > > Furthermore, we will continue to expand our evaluation to the longer context datasets (10k, 20k and 30k tokens) and other filing types. We are particularly interested in seeing the results on longer context, as we typically see model performance degrades as context length increases. And as stated in our previous message, we are committed to including full results in the camera-ready version.
> > > > Thank you again for your constructive feedback and hope this helps address your concerns.

---

> > > > > ### Comment · Reviewer_aMbJ · 2025-08-08
> > > > >
> > > > > I appreciate the detailed response from the authors. I will raise my score.

---

### Official Review · Reviewer_Bnf2 · 2025-07-07

**Rating:** 5
**Confidence:** 3

**Summary:**

The authors propose PHANTOM, which is an open benchmark that hides 500-token “truth chunks” inside 2k–30k-token SEC filings, then asks yes/no questions to see if LLMs can avoid hallucinating when the evidence is far away; results show even 32k-context models crumble once the chunk is ~8k tokens from the start, proving long-context reliability is still unsolved.

**Dataset Code Accessibility:**

No

**Ethical Considerations:**

No, there are no or only very minor ethics concerns

**Final Justification:**

I think the author sufficiently address my concerns.

**Limitations Weaknesses:**

- “SEC docs are public domain” is not a dataset license. Without CC-BY or other licenses, many orgs can’t legally ingest the packaged dataset.
- The author could potentially include more human validation. Only ~4 k seed Qs, and just 10 % got human eyes. That leaves plenty of room for label or question-quality bugs that models might overfit to.
- Modern assistants often juggle multi-doc evidence; PHANTOM sticks to one filing, so retrieval and cross-doc reasoning aren’t measured.
- 80 %+ of errors are “misrepresented numbers.” Good for finance, but weak coverage of other hallucination modes (entity swaps, causal claims, etc.).

**Strengths Contributions:**

- The design is relatively clear. The 500-token “truth chunk” stays the same while total context grows and its position shifts → isolates “lost-in-the-middle” failures instead of mixing in retrieval or reasoning noise.
- The authors use raw SEC filings, which are real-world high stake text.
- The paper is general well-written.

---

> ### Author Rebuttal · Authors · 2025-07-31
>
> We sincerely thank you for your thoughtful evaluation and constructive feedback on our paper. We are encouraged that you found our work to be "technically solid" and that you rated it as a "borderline accept." Your specific concerns have helped us identify concrete improvements that will strengthen both the immediate contribution and long-term impact of PHANTOM.
>
> ## Regarding your comment on whether SEC documents are public domain:
> The source documents, SEC filings, are indeed public domain materials freely accessible via the SEC’s EDGAR database, as they are required regulatory disclosures by public companies. This means anyone can legally access, view, and use the original filings. Please see https://www.sec.gov/about/webmaster-frequently-asked-questions
>
> ## Regarding the licensing of the dataset for organizations:
> In case of acceptance the dataset will be released under Apache 2.0 license, which allows commercial and non-commercial usage. For details visit: https://www.apache.org/licenses/LICENSE-2.0
>
> ## Regarding Human Validation:
> We acknowledge that more extensive human validation would be ideal, our current approach aligns with established practices in the field. The use of synthetic data generation, subsequently validated with a smaller human-annotated set, is a common methodology in recent large-scale benchmarks. This practice is exemplified by works such as HaluEval (Li et al. (2023)) annotated 14.2% and RAGEval (Zhu et al. (2024)) annotated 9.3%. In adherence to these standard practices, we further enhanced our human validation efforts on the seed data. Specifically, we increased the human review coverage from 10% to 38.4% of the seed dataset. In total we reviewed 1,543 out of 3,962 samples, with 21 being labeled as incorrect. Within this expanded review, 98.6% of the answers were confirmed to be correct. With the increased number of samples we still observe accuracies that are comparable to the numbers reported in the paper. We will update the number for the final version of the paper (if accepted).
>
> Li et al. (2023). “HaluEval: A Large-Scale Hallucination Evaluation Benchmark for Large Language Models”. EMLP.
>
> Zhu, K et al. (2024). “RAGEval: Scenario Specific RAG Evaluation Dataset Generation Framework”. CoRR abs/2408.01262.
>
> ## Regarding multi-doc evidence and cross-document retrieval:
> Our work focuses on assessing the impact of long documents, which is a distinct challenge from processing multiple documents (Levy et al. (2025)). We believe that the performance of LLMs on long documents is not well understood, thus adding an additional confounder that can influence performance, i.e., multi-document retrieval, could have diluted the performance measurement. As mentioned in our limitation section, we leave exploring the multi-document scenario for future work. However, we do acknowledge this exploration to be an important extension to the proposed methodology.
>
> Levy et al. (2025). "More Documents, Same Length: Isolating the Challenge of Multiple Documents in RAG," arXiv Preprint.
>
> ## Regarding most hallucinations are categorized as “misrepresented numbers”:
> We assume that the reviewer is referring to Class 2: Misrepresentation, which makes up 81.3% of our samples in the dataset. We wanted to clarify that this class is not restricted to misinterpreted numbers, as it rather reflects arguably the most nuanced and challenging type of hallucination. The following are some instances of class 2 hallucinations that exemplify our argument:
>
> ### Example1:
> Query: What percentage of the company's net revenue was generated outside the United States in 2023?
> Relevant snippet from Context containing the answer: A significant currency fluctuation between the U.S. dollar and other currencies could adversely impact our operating income. With our operations in the Americas, Europe and Asia Pacific, we conduct business using various currencies. In 2023, approximately 41% of our net revenue was generated outside the United States. We do not enter into hedging transactions relating to our exposure to currency fluctuations. As we typically transact business in the local currency of our subsidiaries, our profitability may be impacted by the translation of foreign currency financial statements into U.S. dollars. Significant long-term fluctuations in relative currency values, in particular an increase in the value of the U.S. dollar against foreign currencies, could have an adverse effect on our financial condition and results of operations. Currency fluctuations positively impacted our net revenues 0.3% and negatively impacted our operating income by 0.7% for the year ended December 31, 2023. Our ability to access additional credit could be limited, which may negatively impact our business.
> Hallucinated Answer: In 2023, the company generated approximately 41% of its net revenue outside the United States, primarily driven by operations in the Asia Pacific region, which has been a key focus area for the company's international expansion strategy, leveraging local currencies to mitigate the impact of significant currency fluctuations on profitability.
>
> Analysis:
> The provided answer is classified as "Class 2 Misrepresentation" because it introduces significant information that is not present in the supplied context. Misrepresentation in this sense involves adding details or making assertions that cannot be substantiated by the source material. It accurately states the 41% revenue figure but then incorrectly identifies the Asia Pacific region as the primary driver and falsely reframes a documented currency risk as a successful mitigation strategy. This embellishment misleads the reader about what the provided text actually contains.
>
> ### Example 2:
> Query: What percentage of the Company's total purchases were accounted for by its two largest vendors during the years ended December 31, 2023 and 2022?
> Relevant snippet from Context containing the answer: The Company has not experienced any losses related to its cash and cash equivalents and does not believe that it is subject to unusual credit risk beyond the normal credit risk associated with commercial banking relationships. At December 31, 2023 and 2022, all of the Company's cash and cash equivalents were held at one accredited financial institution. Concentrations The Company is dependent on third-party vendors for its inventory purchases. During the years ended December 31, 2023 and 2022, two vendors accounted for 35% and 59% of total purchases, respectively. The loss of these vendors may have a negative short-term impact on the Company’s operations; however, the Company believes there are acceptable substitute vendors that can be utilized longer-term. Cash and Cash Equivalents The Company considers all highly liquid investments with maturities of three months or less at the date of purchase to be cash equivalents.
> Hallucinated Answer: During the years ended December 31, 2023 and 2022, the Company's two largest vendors accounted for approximately 35% and 59% of total purchases, respectively, with these vendors primarily supplying inventory for the Company's core operations, which has been a consistent trend over the past few years.
>
> Analysis: The answer is classified as "Class 2 Misrepresentation" because it is partially based on the context, but it made an assertion that was not supported by the provided context.
> The answer accurately reported that two vendors accounted for 35% and 59% of total purchases in 2023 and 2022, which was directly supported by the provided financial context. However, the statement that this represented a "consistent trend over the past few years" was a misrepresentation, as the given context only provided data for those two specific years and lacked the historical information necessary to substantiate such a claim of consistency.
>
> ## Regarding Dataset Code Accessibility:
> The dataset and code is available since the time of submission of the paper. Please refer to the guidance in the supplemental material on how to access the dataset.

---

### Note · Authors · 2025-08-15

We sincerely thank the reviewers and the Area Chair for their time and insightful engagement. This discussion has been invaluable in strengthening our work.

Our paper introduces PHANTOM, the first large-scale, publicly available benchmark for evaluating hallucination detection methods in the domain of long-context financial question-answering. This work addresses a critical and previously unmet need for a comprehensive dataset in this high-stakes domain, with the unique feature of enabling fair and direct comparisons of detection performance sensitivity to context length and information placement.

The engaging reviewer discussions gave us the opportunity to significantly improve the paper. Specifically, we have:
1. Expanded our evaluation to include leading closed-source models (including GPT and Gemini families) and more open source models. We have already provided preliminary results and are committed to including a comprehensive evaluation across all long-context settings in the final version.

2. Increased our human validation coverage from 10% to 38.4% of the seed data, substantially reinforcing the quality and reliability of our dataset.

In conclusion, we believe PHANTOM makes a critical contribution to the discourse around responsible AI. By providing a challenging, reproducible, and publicly accessible benchmark, we empower the research community to rigorously test and understand hallucination detection methods in finance, ultimately fostering the development of more trustworthy AI systems.
We thank you for your consideration.

---

### Decision · Program_Chairs · 2025-09-18

**Decision:**

Accept (poster)

**Comment:**

The paper introduces a very interesting and useful benchmark for the important problem of hallucination detection in long-context financial question-answering. Reviewers initially pointed out several key weaknesses regarding the percentage of human data validation and the diversity of the models used in experiments. Authors did an excellent job in addressing all these concerns. Specifically authors substantially increased human validation to over 38% and added new evaluation results based on closed-source models. I believe these detailed clarifications addressed all major concerns. Therefore, I recommend Acceptance.

===== FINAL UPDATE FROM DB Track PCs ====

The final decision for this paper has been taken by the program chairs after consultation with the SACs. All Senior Area Chairs have ranked papers according to the feedback from the AC during the review process. We decided to leave the original meta-review to reflect the opinion of the AC in light of the initial discussions with reviewers and SAC.